# Task-oriented Time Series Imputation Evaluation via Generalized Representers

**Zhixian Wang**[1,2]**, Linxiao Yang**[2]**, Liang Sun**[2]**, Qingsong Wen**[2]**, Yi Wang**[1]*

[1]The University of Hong Kong, [2]DAMO Academy, Alibaba Group
`zxwang@eee.hku.hk`, `linxiao.ylx@alibaba-inc.com`, `liang.sun@alibaba-inc.com`,
`qingsongedu@gmail.com`, `yiwang@eee.hku.hk`

## Abstract

Time series analysis is widely used in many fields such as power energy, economics, and transportation, including different tasks such as forecasting, anomaly detection, classification, etc. Missing values are widely observed in these tasks, and often leading to unpredictable negative effects on existing methods, hindering their further application. In response to this situation, existing time series imputation methods mainly focus on restoring sequences based on their data characteristics, while ignoring the performance of the restored sequences in downstream tasks. Considering different requirements of downstream tasks (e.g., forecasting), this paper proposes an efficient downstream task-oriented time series imputation evaluation approach. By combining time series imputation with neural network models used for downstream tasks, the gain of different imputation strategies on downstream tasks is estimated without retraining, and the most favorable imputation value for downstream tasks is given by combining different imputation strategies according to the estimated gain. The corresponding code can be found in the repository `https://github.com/hkuedl/Task-Oriented-Imputation`.

## 1 Introduction

Time series analysis plays a crucial role in many real-world applications, such as energy, finance, healthcare, and other fields [1, 2, 3]. For example, forecasting load series forms the basis for further decision-making in power dispatch in the power grid system, thereby generating a significant amount of economic benefits [4, 5, 6]. However, collecting time series data, especially high-quality ones, is challenging. Due to the instability of the external environment, sensor failures, and even ethical and legal privacy issues, missing values are prevalent in time series data [7]. For instance, in the BDG2 load series dataset [8], widely used in building energy analysis, the ratio of complete time series data is less than 10%.

To handle missing values in time series data, numerous methods have been proposed for time series imputation in the literature. Based on the features of the imputation methods, these approaches can be divided into statistical and machine learning methods, such as ARIMA and KNN [9, 10], as well as deep learning-based methods [11, 12, 13, 14]. Both types of methods generally use reconstruction errors of missing values to guide learning and perform evaluation. Recently, some researchers have turned their attention to evaluation strategies based on downstream task performance [15]. However, in most cases, downstream tasks are classification tasks [13], while forecasting tasks, as another important branch of time series-related tasks, have not been fully considered. The main challenge for time series forecasting is that the time series serves as both input and label (output) for the model during training, whereas in classification tasks, it only serves as input for the model.

---

*Corresponding author.

38th Conference on Neural Information Processing Systems (NeurIPS 2024).

In supervised learning, training labels influence the calculation of the loss function, which in turn affects the optimization of model parameters and, ultimately, the performance of the model on the test set. [16] indicates that noise in input data (missing data can be considered a type of noise) often has a limited impact on forecasting results. In contrast, label noise can significantly affect the model and, consequently, the final test results from the beginning to the end of the time series. Therefore, when evaluating the impact of different time imputation methods on downstream forecasting tasks, it is essential to focus on assessing the quality of training labels constructed through various imputation methods.

To evaluate how the quality of the imputation labels affects downstream forecasting tasks, it is important to clarify that an excellent imputation strategy does not necessarily mean that the imputed value at each time step is superior to any other method. [15] provides a benchmark for various methods in time series imputation tasks. Although SAITS [13], as one of the latest SOTA methods, has achieved remarkable results, there are still methods that surpass SAITS in some cases. This demonstrates that time series imputation is a complex task, making it difficult to find a universal method capable of handling all situations, let alone considering the performance of downstream forecasting tasks. A more realistic scenario is that while one method may perform better overall, it may not outperform other methods locally. In time series, this means that one method may excel in some time steps, while others do better in different time steps. However, to examine the impact of each time step on forecasting, retraining the forecasting model multiple times is necessary, which is impractical due to time and computational costs. Therefore, an efficient estimation method is needed to examine the impact of each time step with different imputation methods. Additionally, since finding a universal method is difficult, it is natural to shift focus toward combining the advantages of current methods to obtain a better imputation strategy. Consequently, determining how to combine different strategies becomes a challenge.

Based on the above situation, we have developed a task-oriented time series imputation evaluation strategy. Specifically, we summarize our contributions into the following points.

1. We propose a strategy that evaluates each time series imputation method by estimating the impact of missing (imputed) labels at each time step on downstream tasks without requiring multiple retraining, which significantly reduces time and computational consumption. To the best of our knowledge, we are the first to consider the impact of missing values in time series as labels on downstream forecasting tasks.

2. We introduce a simple and effective similarity calculation method based on the characteristics of long time series to estimate the impact of imputed values more quickly, striking a balance between performance and computational cost.

3. We develop a time series imputation framework guided by maximizing the gains of downstream tasks, enabling the combination of advantages from different time series imputation strategies to achieve a better one. This results in improved performance in the downstream forecasting task.

## 1.1 Related Work

**Time Series Imputation** Time series imputation can be primarily classified into two categories: traditional techniques and neural network-based techniques. Traditional methods replace missing values with statistics, such as the mean value or the last observed value [17]. They include simple statistical models like ARIMA [18], ARFIMA, SARIMA [19], and machine learning techniques such as KNNI [20], TIDER [21], MICE [22], BayOTIDE [23]. In recent years, deep learning imputation methods have demonstrated remarkable capabilities in capturing intricate temporal relationships and complex variation patterns inherent in time series data. These methods employ deep learning models like Transformers [24, 13], generative neural networks such as VAEs [12, 25], GANs [26, 27], and diffusion models [28] to capture complex dynamic relationships within time series data. Although different methods exhibit various advantages, no universal method currently outperforms others in all scenarios and datasets. This observation inspires us to consider combining existing advanced methods in this work to achieve better time series imputation strategies.

**Sample-based Explaination** Sample-based explainable methods can be divided into two categories [29]. One is based on retraining, which evaluates the importance of corresponding data by comparing the impact of removing data points on the model and even the final test results [30, 31, 32].

Among them, the introduction of shapley value by [30] naturally ensures the fairness of data attribution. The other type is based on gradient methods, which directly estimate the influence of data points without the need for retraining. This type of method can be subdivided into three main categories, which are based on representative theories, Influence Function, and training loss trajectory. [33] is a representative of the method of the first type, whose core idea is to fix the other layers and only focus on the last layer of the neural network, so that the influence of each sample can be explicitly calculated. On the other hand, the Influence Function [34] is based on the assumption of convergence and uses the Hessian matrix to estimate the influence of samples. The last type of method takes into account the entire training process of the neural network, continuously tracking the impact of samples on each parameter update [35]. In addition, [36] summarizes the gradient-based method and unifies them as generalized representers.

## 2 Methodology

### 2.1 Problem Statement

Consider a multivariate time series dataset represented by $\{(\boldsymbol{X}_i, \boldsymbol{y}_i)\}_{i=1}^n$, incorporating $n$ samples. In this dataset, $\boldsymbol{X}_i \in \mathbb{R}^{D \times L_1}$ corresponds to a feature matrix containing $D$ distinctive features over $L_1$ temporal intervals, whereas $\boldsymbol{y}_i \in \mathbb{R}^{L_2}$ signifies the target time series, which spans $L_2$ temporal intervals. It is crucial to recognize that $\boldsymbol{y}_i$ may include several missing entries, a common complication within real-world datasets. For example, in the context of electrical load forecasting, $\boldsymbol{X}_i$ encompasses daily weather-related time series data, comprising variables such as temperature, humidity, and wind speed, while $\boldsymbol{y}_i$ represents the electrical load time series of a given day, possibly containing missing entries due to issues in data collection or transmission.

Addressing missing values in $\{\boldsymbol{y}_i\}$ through imputation is a fundamental preprocessing step for machine learning tasks involving this data, underscoring the necessity to assess the effectiveness of various imputation methods. Consider $\{\boldsymbol{y}_i^{(1)}\}$ and $\{\boldsymbol{y}_i^{(2)}\}$ as two time series resulting from the imputation of $\{\boldsymbol{y}\}$ via two different methods. The goal is to ascertain whether the imputation performed on $\{\boldsymbol{y}_i^{(2)}\}$ is superior to that on $\{\boldsymbol{y}_i^{(1)}\}$. Moreover, we seek to evaluate the quality of imputation at each temporal interval, determining if the imputation of the $l$-th interval in $\boldsymbol{y}_i^{(2)}$ is more accurate than that in $\boldsymbol{y}_i^{(1)}$.

Conventionally, the quality of imputation is quantified by measuring the discrepancy between the imputed values and the actual data, favoring methods that minimize this deviation. In this study, however, we propose to assess imputation quality based on the performance of subsequent tasks.

One step further, we evaluate the quality of imputation on a timestep basis, examining if the imputation for the $l$-th interval in $\boldsymbol{y}_i^{(2)}$ exhibits improved efficacy over $\boldsymbol{y}_i^{(1)}$, thereby offering a more nuanced and comprehensive evaluation of imputation methodologies.

Let us define the loss function for the downstream task as $\mathcal{L}(f(\boldsymbol{X}, \boldsymbol{\theta}), \boldsymbol{y})$, where $f(\cdot, \boldsymbol{\theta})$ denotes the model used in the downstream task parametered by $\boldsymbol{\theta}$. And let $\{(\boldsymbol{X}_i^v, \boldsymbol{y}_i^v)\}_{i=1}^m$ constitute a test dataset that will be used to gauge model performance. We denote $y_{i,l}^{(1)}$ and $y_{i,l}^{(2)}$ as the $l$-th entries of $\boldsymbol{y}_i^{(1)}$ and $\boldsymbol{y}_i^{(2)}$, respectively. According to our intuition, if $y_{i,l}^{(2)}$ is superior to $y_{i,l}^{(1)}$, swapping $y_{i,l}^{(1)}$ for $y_{i,l}^{(2)}$ should result in a decrease in the test set's loss. Guided by this rationale, we define the indicator function $I(i, l)$, which discerns whether $y_{i,l}^{(2)}$ is preferable over $y_{i,l}^{(1)}$ as follows:

$$I(i,l) = \sum_{k=1}^m I(i,l,\boldsymbol{X}_k^v) = \sum_{k=1}^m \left( \mathcal{L}(f(\boldsymbol{X}_k^v, \boldsymbol{\theta}_1), \boldsymbol{y}_k^v) - \mathcal{L}(f(\boldsymbol{X}_k^v, \boldsymbol{\theta}_2), \boldsymbol{y}_k^v) \right)$$

$$\text{s.t. } \boldsymbol{\theta}_1 = \arg\min_{\boldsymbol{\theta}} \sum_{k=1}^n \mathcal{L}(f(\boldsymbol{X}_k, \boldsymbol{\theta}), \boldsymbol{y}_k^{(1)})$$

$$\boldsymbol{\theta}_2 = \arg\min_{\boldsymbol{\theta}} \mathcal{L}(f(\boldsymbol{X}_i, \boldsymbol{\theta}), \overline{\boldsymbol{y}_i}^{(2,l)}) + \sum_{k \neq i}^n \mathcal{L}(f(\boldsymbol{X}_k, \boldsymbol{\theta}), \boldsymbol{y}_k^{(1)}). \tag{1}$$

Here, $\overline{\boldsymbol{y}_i}^{(2,l)}$ denotes a vector identical to $\boldsymbol{y}_i^{(1)}$, except at the $l$-th entry, which matches that of $\boldsymbol{y}_i^{(2)}$. Clearly, $I(i,l) \geq 0$ implies that the substitution of $y_{i,l}^{(1)}$ with $y_{i,l}^{(2)}$ leads to a decreased test set loss, suggesting that $y_{i,l}^{(2)}$ is superior. Conversely, if $I(i,l) < 0$, it suggests that $y_{i,l}^{(1)}$ is preferable to $y_{i,l}^{(2)}$.

Despite the effectiveness of the definition provided by Equation (1), computing $I(i,l)$ for every missing value in the dataset is impractical due to the extensive model retraining required, which can be prohibitive in terms of time. To overcome this challenge, in the next section, we put forth an efficient methodology for estimating $I(i,l)$ without retraining the model.

## 2.2 Approximation Model Construction

To compute $I(i,l)$ efficiently, we propose a retrain-free method in this subsection. As both $\boldsymbol{y}^{(1)}$ and $\boldsymbol{y}^{(2)}$ are imputation of $\boldsymbol{y}$, then we assume that $y_{i,l}^{(1)}$ is close to $y_{i,l}^{(2)}$, with which we approximate $I(i,l)$ using the first order Tolyer expansion as

$$
\begin{aligned}
I(i,l) &\approx \sum_{k=1}^{m} \left.\frac{\partial \mathcal{L}(f(\boldsymbol{X}_k^v, \boldsymbol{\theta}), \boldsymbol{y}_k^v)}{\partial y_{i,l}}\right|_{y_{i,l}=y_{i,l}^{(1)}} (y_{i,l}^{(1)} - y_{i,l}^{(2)}) \\
&= \sum_{k=1}^{m} \left.\frac{\partial \mathcal{L}(f(\boldsymbol{X}_k^v, \boldsymbol{\theta}), \boldsymbol{y}_k^v)}{\partial f(\boldsymbol{X}_k^v, \boldsymbol{\theta})}^T \frac{\partial f(\boldsymbol{X}_k^v, \boldsymbol{\theta})}{\partial y_{i,l}}\right|_{y_{i,l}=y_{i,l}^{(1)}} (y_{i,l}^{(1)} - y_{i,l}^{(2)}).
\end{aligned}
\tag{2}
$$

Equation (2) provides an approximation for computing $I(i,l)$, where $\frac{\partial f(\boldsymbol{X}_k^v, \boldsymbol{\theta})}{\partial y_{i,l}}$ measures how the training target $y_{i,l}$ affect the prediction of the test data, and $\frac{\partial \mathcal{L}(f(\boldsymbol{X}_k^v, \boldsymbol{\theta}), \boldsymbol{y}_k^v)}{\partial f(\boldsymbol{X}_k^v, \boldsymbol{\theta})}$ computes how the changing of the prediction of $\boldsymbol{X}_k^v$ affect the final loss. Note that the symbolic expression $\frac{\partial f(\boldsymbol{X}_k^v, \boldsymbol{\theta})}{\partial y_{i,l}}$ can be conceptually broken down into $\frac{\partial f(\boldsymbol{X}_k^v, \boldsymbol{\theta})}{\partial \boldsymbol{\theta}} \cdot \frac{\partial \boldsymbol{\theta}}{\partial y_{i,l}}$, elucidating the role of the label $y_{i,l}$ in shaping the model parameters $\boldsymbol{\theta}$ throughout the training process. This, in turn, has repercussions on the model's prediction when evaluated on unseen data from the test set, i.e. $f(\boldsymbol{X}_k^v, \theta)$. By focusing on the derivative $\frac{\partial f(\boldsymbol{X}_k^v, \boldsymbol{\theta})}{\partial y_{i,l}}$, our goal is to assess the extent to which changes in label values $y_{i,l}$ influence the model's predictions on the test set, thereby affecting overall model efficacy.

When it comes back to the estimation, dispite $\frac{\partial \mathcal{L}(f(\boldsymbol{X}_k^v, \boldsymbol{\theta}), \boldsymbol{y}_k^v)}{\partial f(\boldsymbol{X}_k^v, \boldsymbol{\theta})}$ and $y_{i,l}^{(1)} - y_{i,l}^{(2)}$ are easy to compute, estimating $\frac{\partial f(\boldsymbol{X}_k^v, \boldsymbol{\theta})}{\partial y_{i,l}}$ is difficult. The difficulty comes from two aspects. Firstly, for the complex $f(\cdot, \boldsymbol{\theta})$, the final parameter is not only affected by the training data, some other factors, such as the structure of the network and learning rate during the learning process. Secondly, all of the $n$ training samples affect the parameters of the model, leading to the mixture of the effect of data points on the final model. Thus isolating the effect of a single data point is difficult.

To overcome these two difficulties, we propose to approximate $\frac{\partial f(\boldsymbol{X}_k^v, \boldsymbol{\theta})}{\partial y_{i,l}}$ using a white-box model, where how each training datapoint affects the final prediction is clear from the design of the model. To this end, we propose to approximate $\frac{\partial f(\boldsymbol{X}_k^v, \boldsymbol{\theta})}{\partial y_{i,l}}$ using a kernel machine, i.e. $\boldsymbol{\alpha}_{i,l}^T K(\boldsymbol{X}_i, \boldsymbol{X}_k^v)$, where $K(\boldsymbol{X}_i, \cdot)$ is a kenerl between the training sample $\boldsymbol{X}_i$ and test samples measuring the similarity between the $\boldsymbol{X}_i$ and $\boldsymbol{X}_k^v$, and $\boldsymbol{\alpha}$ is a learnerable hyperparameter. It can be proven that the indicator function based on this definition satisfies many desirable properties (please see Appendix for details) to construct an axiomatic attribution. Formally, the coefficient $\boldsymbol{\alpha_{i,l}} \in \mathbb{R}^{L_2}$ can be computed by solving the following optimization problem:

$$
\hat{\boldsymbol{\alpha}} = \underset{\boldsymbol{\alpha} \in \mathbb{R}^n \times \mathbb{R}^{L_2} \times \mathbb{R}^{L_2}}{\operatorname{argmin}} \left\{ \sum_{i=1}^{n} \sum_{l=1}^{L_2} \sum_{j=1}^{n} \mathcal{L}\left(\boldsymbol{\alpha}_{i,l}^T K(\boldsymbol{X}_i, \boldsymbol{X}_j), \frac{\partial f(\boldsymbol{X}_j, \boldsymbol{\theta})}{\partial y_{i,l}}\right) \right\}.
\tag{3}
$$

To solve this problem, $\frac{f(\boldsymbol{X}_j, \boldsymbol{\theta}_1) - f(\boldsymbol{X}_j, \boldsymbol{\theta}_2)}{y_{i,l}^{(1)} - y_{i,l}^{(2)}}$ can act as a substitute of $\frac{\partial f(\boldsymbol{X}_j, \boldsymbol{\theta})}{\partial y_{i,l}}$ since there is no ground truth. However, the problem is still not practical to solve because we can not obtain $f(\boldsymbol{X}_j, \boldsymbol{\theta}_2)$ without retraining the model. Even though it isn't necessary to traverse all $i$, $j$, and $l$ to obtain the

complete data, it still goes against our original intention to calculate $I(i, l)$ efficiently. Furthermore, it is also difficult for us to determine how much data is sufficient to ensure the accuracy of the solution. Fortunately, with the help of Remark 1, we can bypass the process of finding enough $\frac{f(\boldsymbol{X}_j, \boldsymbol{\theta}_1) - f(\boldsymbol{X}_j, \boldsymbol{\theta}_2)}{y_{i,l}^{(1)} - y_{i,l}^{(2)}}$ and directly approximate $\frac{\partial f(\boldsymbol{X}_j, \boldsymbol{\theta})}{\partial y_{i,l}}$.

**Remark 1.** *Given two infinitely differentiable functions $f(\boldsymbol{x})$ and $g(\boldsymbol{x})$ in a bounded domain $D \in R^n$, $||f(\boldsymbol{x}) - g(\boldsymbol{x})||$ is always less than $\epsilon$. For any given $\delta$ and $\epsilon_2$, there exists an $\epsilon$ such that, in the domain $D$, the measure of the region $I$ that satisfying $||\frac{\partial f(\boldsymbol{x})}{\partial \boldsymbol{x}} - \frac{\partial g(\boldsymbol{x})}{\partial \boldsymbol{x}}|| > \delta$ is not greater than $\epsilon_2$, i.e, $m(I) \leq \epsilon_2$.*

Remark 1 gives us the intuition that we can avoid retraining our downstream models by first-order approximation. However, some issues still need to be clarified. First, we consider neural networks used in the downstream task as infinitely differentiable functions since in practical applications, it is unlikely for computed floating-point numbers to precisely equal non-differentiable points. Second, Remark 1 limits the definition domain to a bounded region $D$. Time series data is usually bounded (for example, the renewable generation sequence cannot be greater than the installed capacity), making this assumption reasonable. Finally, Remark 1 can be rephrased as the better the approximation of the original function, the better the approximation of its derivative, that is, we can have a $\frac{m(D) - \epsilon_2}{m(D)}$ probability of fitting the derivative well. Therefore, the optimization problem (3) can be transformed into an easier one. (Note that we indicate the existence of $\epsilon$ that meets the conditions in this remark, but make no restrictions on $\epsilon$, while it is often difficult for us to make the approximation error of the original function sufficiently small. If the $\epsilon$ in the remark is infinitely close to 0, practical applications will encounter difficulties. However, in gradient descent-based neural network training, such situations often do not hinder our practical applications. Due to the space limit, the full theoretical discussion is provided in the Appendix.)

$$\hat{\boldsymbol{\alpha}}' = \underset{\boldsymbol{\alpha}' \in \mathbb{R}^n \times \mathbb{R}^{L_2}}{\operatorname{argmin}} \left\{ \sum_{i=1}^n \mathcal{L} \left( \sum_{j=1}^n \boldsymbol{\alpha}'^T_j K(\boldsymbol{X}_i, \boldsymbol{X}_j), f(\boldsymbol{X}_i, \boldsymbol{\theta}) \right) \right\}, \tag{4}$$

$$\hat{\boldsymbol{\alpha}}_{i,l} = \frac{\partial \hat{\boldsymbol{\alpha}}'_i}{\partial y_{i,l}}. \tag{5}$$

Now the problem is converted to solving the problem (4). Intuitively, we solve it by projecting it onto the RKHS subspace spanned by the kernels,

$$\hat{\boldsymbol{\alpha}}' = \underset{\boldsymbol{\alpha}' \in \mathbb{R}^n \times \mathbb{R}^{L_2}}{\operatorname{argmin}} \left\{ \sum_{i=1}^n \mathcal{L} \left( \underbrace{\sum_{j=1}^n \boldsymbol{\alpha}'^T_j K(\boldsymbol{X}_i, \boldsymbol{X}_j)}_{f_K(\boldsymbol{X}_i)}, f(\boldsymbol{X}_i, \boldsymbol{\theta}) \right) + \underbrace{\frac{1}{2} \sum_{l=1}^{L_2} \boldsymbol{\alpha}'^\top_{.,l} \boldsymbol{K}_l \boldsymbol{\alpha}'_{.,l}}_{\|f_K\|^2_{\mathcal{H}_K}} \right\}, \tag{6}$$

where $\boldsymbol{K}_l$ is the kernel gram matrix defined as $\boldsymbol{K}_{l,ij} = K(\boldsymbol{X}_i, \boldsymbol{X}_j)_l$. Considering the first-order optimality condition, $\hat{\alpha}'_{i,l} = -\frac{1}{n} \frac{\partial \mathcal{L}(\hat{f}_K(\boldsymbol{X}_i), f(\boldsymbol{X}_i, \boldsymbol{\theta}))}{\partial \hat{f}_K(\boldsymbol{X}_i)_l}$ [36, 37]. Recalling our goal of estimating the relationship between $\hat{\boldsymbol{\alpha}}'$ and $y_{i,l}$ in (5), their relationship is still unclear. This is because the objective of the optimization problem is to construct an approximation model without considering the role of label value $y_{i,l}$. To clarify the effect of $y_{i,l}$, we introduce an approximation by trigonometric inequality that $\mathcal{L}(f_K(\boldsymbol{X}_i), f(\boldsymbol{X}_i, \boldsymbol{\theta})) \leq \mathcal{L}(\boldsymbol{y}_i, f(\boldsymbol{X}_i, \boldsymbol{\theta})) + \mathcal{L}(f_K(\boldsymbol{X}_i), \boldsymbol{y}_i)$. The second term on the right side is the loss function corresponding to the downstream model, and in the case of training convergence, this should be close to a constant. Therefore, the optimization problem (6) can be rewritten as

$$\hat{\boldsymbol{\alpha}}' = \underset{\boldsymbol{\alpha}' \in \mathbb{R}^n \times \mathbb{R}^{L_2}}{\operatorname{argmin}} \left\{ \sum_{i=1}^n \left( \mathcal{L}(\boldsymbol{y}_i, f(\boldsymbol{X}_i, \boldsymbol{\theta})) + \mathcal{L}(f_K(\boldsymbol{X}_i), \boldsymbol{y}_i) \right) + \underbrace{\frac{1}{2} \sum_{l=1}^{L_2} \boldsymbol{\alpha}'^\top_{.,l} \boldsymbol{K}_l \boldsymbol{\alpha}'_{.,l}}_{\|f_K\|^2_{\mathcal{H}_K}} \right\}. \tag{7}$$

Formally, the solution are $\hat{\boldsymbol{\alpha}}_i' = -\frac{1}{n} \frac{\partial \mathcal{L}(\hat{f}_K(\boldsymbol{X}_i), \boldsymbol{y}_i)}{\partial \hat{f}_K(\boldsymbol{X}_i)}$ and $\hat{\boldsymbol{\alpha}}_{i,l} = -\frac{1}{n} \frac{\partial^2 \mathcal{L}(\hat{f}_K(\boldsymbol{X}_i), \boldsymbol{y}_i)}{\partial \hat{f}_K(\boldsymbol{X}_i) \partial y_{i,l}}$. Furthermore, since $f_K(\cdot)$ is used to approximate $f(\cdot, \boldsymbol{\theta})$, we use $f(\cdot, \boldsymbol{\theta})$ to replace $f_K(\cdot)$ to simplify the calculation. With the above preparation, the $I(i, l)$ can be represented as follow with NTK kernel [38],

$$\sum_{k=1}^{m} -\frac{1}{n} \frac{\partial \mathcal{L}(f(\boldsymbol{X}_k^v, \boldsymbol{\theta}), \boldsymbol{y}_k^v)}{\partial f(\boldsymbol{X}_k^v, \boldsymbol{\theta})} \underbrace{\frac{\partial^2 \mathcal{L}(f(\boldsymbol{X}_i, \boldsymbol{\theta}), \boldsymbol{y}_i)^T}{\partial f(\boldsymbol{X}_i, \boldsymbol{\theta}_t) \partial y_{i,l}}}_{\hat{\boldsymbol{a}_{i,l}}} \underbrace{\frac{\partial f(\boldsymbol{X}_i, \boldsymbol{\theta})}{\partial \boldsymbol{\theta}} \frac{\partial f(\boldsymbol{X}_k^v, \boldsymbol{\theta})^T}{\partial \boldsymbol{\theta}}}_{NTK\,kernel} . \quad (8)$$

## 2.3 Similarity Calculation Acceleration

In the previous section, to gauge the effects of substituting $y_{i,l}^{(1)}$ with $y_{i,l}^{(2)}$ on the downstream task, we utilized the Neural Tangent Kernel to assess the similarity between the model outputs for inputs $\boldsymbol{X}_i$ and $\boldsymbol{X}_k^v$. Given that the model's output length is $L_2$, the computational complexity of calculating $I(i, l)$ for all time steps in $\boldsymbol{y}$ scales as $\mathcal{O}(mL_2P)$, where $P$ denotes the total number of parameters in the model $f(\cdot, \boldsymbol{\theta})$, i.e., $|\boldsymbol{\theta}|$. In numerous time series applications, such as forecasting, $L_2$ can be substantially large (e.g., 128), rendering the evaluation process for all imputations time-consuming. To mitigate this challenge, we propose a method to compress the size of $\frac{\partial f(\boldsymbol{X}_i, \boldsymbol{\theta})}{\partial \boldsymbol{\theta}}$. Our approach is inspired by the observation that in time series forecasting, the output of $f(\cdot, \boldsymbol{\theta})$ typically exhibits smooth variability across different $l$ values. Therefore, we posit that the model output $f(\boldsymbol{X}_i, \boldsymbol{\theta})$ resides in a low-dimensional space spanned by a limited number of smooth basis functions. In mathematical terms, $f(\boldsymbol{X}_i, \boldsymbol{\theta}) \approx \boldsymbol{A}^\dagger \boldsymbol{A} f(\boldsymbol{X}_i, \boldsymbol{\theta})$ where $\boldsymbol{A} \in \mathbb{R}^{r \times L_2}$ consists of rows each representing a predefined smooth vector, $\boldsymbol{A}^\dagger \in \mathbb{R}^{L_2 \times r}$ is the pseudo-inverse of $\boldsymbol{A}$, and $r$, which is significantly smaller than $L_2$, denotes the number of basis functions employed to approximate $f(\boldsymbol{X}_i, \boldsymbol{\theta})$. Consequently, we can approximate $\frac{\partial f(\boldsymbol{X}_i, \boldsymbol{\theta})}{\partial \boldsymbol{\theta}}$ as:

$$\frac{\partial f(\boldsymbol{X}_i, \boldsymbol{\theta})}{\partial \boldsymbol{\theta}} \approx \boldsymbol{A}^\dagger \frac{\partial \boldsymbol{A} f(\boldsymbol{X}_i, \boldsymbol{\theta})}{\partial \boldsymbol{\theta}}. \quad (9)$$

This approximation allows for the compression of the model output, thereby reducing the number of gradients that require computation. Through this simplification, the computational complexity for calculating $I(i, l)$ decreases to $\mathcal{O}(mrP)$, substantially less than the original complexity.

In our experiments, we further simplify by assuming $\boldsymbol{A}$ to be a block diagonal matrix, defined as blkdiag$(\boldsymbol{1}_1, \boldsymbol{1}_2, \ldots, \boldsymbol{1}_c)$, where $\boldsymbol{1}_1 = \cdots = \boldsymbol{1}_{c-1}$ are vectors of length $\lfloor L_2/c \rfloor$ with all elements equal to 1, and $\boldsymbol{1}_c \in \mathbb{R}^{L_2 - (c-1)\lfloor L_2/c \rfloor}$ is a vector with all elements equal to 1.

## 2.4 Task-oriented Imputation Evaluation

We have introduced a method for computing the indicator $I(i, l)$, which assesses if replacing $y_{i,l}^{(1)}$ with $y_{i,l}^{(2)}$ results in a reduced loss for the downstream task. Given two sets of imputation results, $\{\boldsymbol{y}_i^{(1)}\}_{i=1}^n$ and $\{\boldsymbol{y}_i^{(2)}\}_{i=1}^n$, derived from distinct imputation techniques, we can evaluate $I(i, l)$ across all samples and time steps, and identify that $\boldsymbol{y}_i^{(2)}$ outperforms $\boldsymbol{y}_i^{(1)}$ at time step $l$ if $I(i, l)$ greater than zero and vice versa if the value is lesser. In contrast to conventional evaluation strategies, our proposed method does not necessitate the availability of the ground truth $\boldsymbol{y}$, thereby enhancing its practical utility in myriad real-world scenarios where actual values remain unobtainable. This feature renders our approach significantly more adaptable to situations where empirical truths are elusive.

## 2.5 Task-oriented Imputation Emsemble

Given that our proposed method can evaluate the quality of two imputations at the time step level, a natural extension is to combine these two sets of imputations to derive an improved result. Specifically, we can generate a new set of imputations $\boldsymbol{y}_i'$ for the $i$-th sample, where the $l$-th entry is $y_{i,l}^{(2)}$ if $I(i, l) > 0$, and $y_{i,l}^{(1)}$ otherwise. Based on the definition of $I(i, l)$, we anticipate that a model trained using $\boldsymbol{y}_i'$ will incur a lower loss compared to one trained with $\boldsymbol{y}_i^{(1)}$, thereby yielding better imputation results. It is important to note, however, that the calculation of $I(i, l)$ is predicated

on the scenario where only the $l$th timestep in the $i$th sample from $\{\boldsymbol{y}^{(1)}\}_{i=1}^n$ is substituted with $y_{i,l}^{(2)}$. This consideration omits the potential interactions among samples. Consequently, in practical implementations, we opt to substitute only the timesteps that rank within the top $c\%$ of $I(i,l)$ values. Our experiments, detailed in 3.3.2, confirm the efficacy of our proposed task-oriented imputation ensemble method. Our proposed ensemble process is summarized in Algorithm 1.

---

**Algorithm 1:** Task-oriented Imputation Emsemble.

**Data:** Training data $\{(\boldsymbol{X}_i, \boldsymbol{y}_i^{(1)})\}_n$ and $\{(\boldsymbol{X}_i, \boldsymbol{y}_i^{(2)})\}_n$; Estimated gain on Validation data $\{I(i,l)\}_{i=1:n,l=1:L_2}$; Downstream task model $f(\cdot, \boldsymbol{\theta})$; Replacing percentage $c\%$

1   Calculate $threshold = P_{1-c}(\{I(i,l)|I(i,l) > 0\})$
2   **for** $i = 1$ *to* $n$, $l = 1$ *to* $L_2$ **do**
3      **if** $I(i,l) > threshold$ **then**
4         Let $y'_{i,l} = y_{i,l}^{(2)}$
5      **else**
6         Let $y'_{i,l} = y_{i,l}^{(1)}$
7      **end**
8   **end**
9   Train $f(\cdot, \boldsymbol{\theta})$ on $\{(\boldsymbol{X}_i, \boldsymbol{y}'_i)\}_n$
   **Result:** $f(\cdot, \boldsymbol{\theta})$

---

# 3 Experiment

## 3.1 Datasets and Experiment Setup

To validate our method, we conduct experiments on six datasets: the GEF load forecasting competition dataset with the corresponding temperature [39], the UCI dataset (electricity load and air quality) [40], the Traffic dataset containing road occupancy rates[2], and two transformer datasets, ETTH1 and ETTH2 [41]. Note that we use the hourly resolution version of the UCI electricity dataset from [42] in the main experiment. In our main experiment, we set the downstream task as univariate time series forecasting, with both input sequence and prediction lengths set to 24. In addition to the GEF dataset, we implement our method on the 'OT' time series in ETH1 and ETH2, the mean value of the road occupancy rate in Traffic, the temperature in the UCI air quality dataset, and the total electricity consumption of 321 users in the UCI electricity dataset. It is important to note that there are no original missing values in these datasets. To simulate the missing values situation, we randomly set masks with lengths in [2, 4, 6, 12, 24, 48, 96, 120], and replace the original values with the average value at the corresponding positions as the baseline. For the missing rate setting, if the missing rate is too low, the difference between different imputation methods may be small, while if the missing rate is too high, the even best imputation method will also be difficult to obtain reasonable results. Based on the above considerations, we mainly consider 40% missing rates as our main experimental setup. Meanwhile, we will provide experimental results under other missing rate settings in [30%, 50%, 60%] in the Appendix.

## 3.2 Time Series Imputation Methods

To verify the performance of our strategy in evaluating different time imputation methods, we introduce multiple advanced time imputation methods. Firstly, as mentioned in the last subsection, we use the basic mean value imputation as the baseline. Secondly, we consider several time series imputation methods based on deep neural networks. They are GPVAE [25] and USGAN [27] based on generative neural network, mRNN [43] and BRITS [11] based on RNN structure, SAITS [13], and ImputeFormer [44] based on attention mechanism. All the implementations of the above models are with the help of the toolkit called PyPOTS [45]. In addition, we also implement the network structure of a spatiotemporal graphs-assisted method called SPIN [46] for time series imputation. All the details of the neural network setting will be described in the Appendix.

---

[2] https://pems.dot.ca.gov

### 3.3 Experimental Results

#### 3.3.1 Application 1: Estimate the Gain

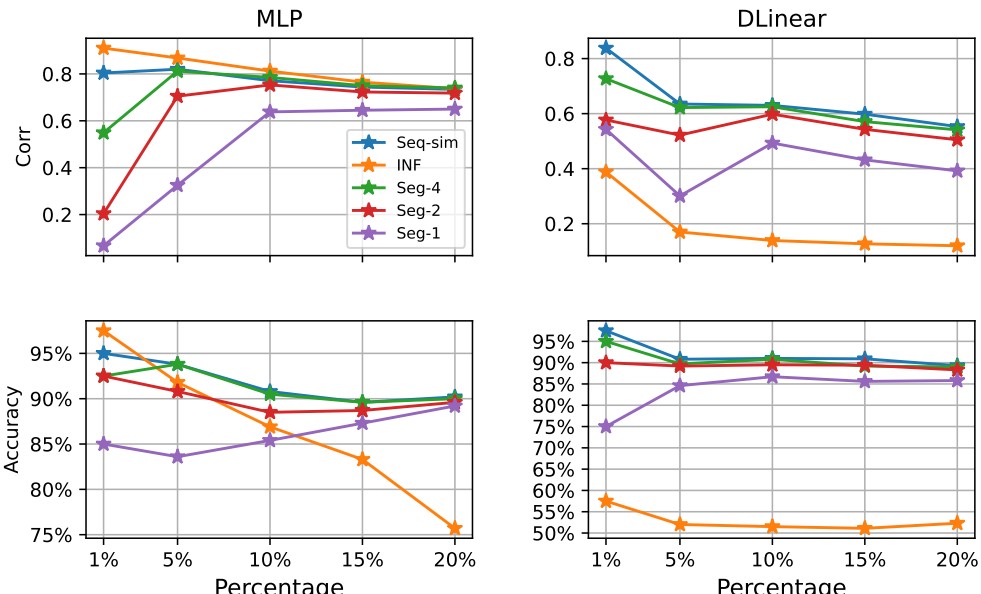

Figure 1: The correlation and accuracy comparison between the estimation of imputation value gain and actual gain (MSE↓), where INF (section D.5) represents our modified Influence Function, Seq-sim represents our original method, and Seg-N represents the acceleration method divided by N segments. The horizontal axis here represents selecting the sample with the highest x% influence based on the absolute value of the estimation.

In this section, we examine the estimated gains of imputation for each time step. We divide the GEF dataset into a training set and a test set, where the training set includes load data from 2011 and the test set includes load data from 2012. We replace 40% of the load data in the training set with the average value at the corresponding position as the baseline, apply the linear interpolation method to replace the corresponding baseline, and use them as training labels separately. In the training set, we have obtained a total of 8760 training samples. Therefore, we replace the labels of each sample one by one to construct new 8760 sets of samples, retrain 8760 forecasting models, test their performance on the test set, and compare the performance of the new model with the model trained on the mean value-based samples. Note that although our estimation is done at each time step of each sample, the time consumption of replacing each time step and retraining is too high. Therefore, we replace all the time steps in each sample and sum up the benefit estimates for each time step as the benefit estimate for the entire sample after replacement. We adopt the 3-layer MLP structure used in [35] and extend it to be used for time series. In addition, we have also added a simple and widely used forecasting model, DLinear [47], as our forecasting backbone model.

Table 1: Time comparison between different methods.

| Time | Seq-sim | Seg-4 | Seg-2 | Seg-1 | Retraining |
|---|---|---|---|---|---|
| MLP | 46s | 18s | 17s | 11s | >48h |
| DLinear | 10s | 8s | 8s | 7s | >48h |

As shown in Figure 1, we use two metrics to compare the performance of different methods, where Corr represents the correlation between estimated gain and actual gain, and accuracy represents the percentage of the same sign between actual gain and estimated gain. In the MLP model, the Influence Function achieves a good correlation. However, in terms of accuracy, the accuracy of the Influence Function rapidly decreases as the percentage of estimated data to total data increases. On the contrary, our method exhibits good characteristics at different percentages. In addition, the

segmented acceleration method generally shows that the more segments there are, the better the performance. Although it performs poorly in small percentages, it exhibits good performance in a wide range of data. In the DLiner model, both our method and the segmented acceleration method present similar situations to those in the MLP model. However, the Influence Function exhibits poor performance, with estimation accuracy and correlation significantly lagging behind our method. In addition, Table 1 reports the total time required for different methods to run on the GTX4090, indicating that retraining requires a significant amount of time while our method achieves a good balance between performance and time consumption. It is worth mentioning that when we focus not only on the small portion of time steps with the greatest impact but also examine the impact of relatively large time steps on the forecasting results, our segmented acceleration method achieves very good performance while reducing the required time.

### 3.3.2 Application 2: Combine Different Time Series Imputation

Similar to the previous section, we use mean value as a baseline to estimate the benefits on the validation set (not test set) obtained by imputation at each time step and replace the $10\%$ time step with the highest benefits to train the forecasting model. In addition, the missing values in the time series damage the original characteristic of the time series. Although imputation values can repair it to some extent, they may still have adverse effects on downstream tasks. Therefore, we also introduce the Influence Function as a comparison, using it to estimate the $10\%$ points with the worst impact on the forecasting results after imputation and remove them, which is often the case in the application of Influence Function [34].

Table 2 reports the comparison of MSE for downstream forecasting tasks (here we use the DLinear as the backbone model), and we will analyze from two aspects.

**I. Comparison between Original and Gain Estimation**. Part I and II in Table 2 demonstrate that combining different imputations can enhance the performance of downstream tasks on all datasets. The improvements in the GEF dataset, Electricity dataset, and Traffic dataset are significant, while the enhancement in others is relatively less noticeable. The primary reason is that, in those datasets, the performance of other imputation methods considerably lags behind the original one, making the replacement of the original label with a newly imputed label seem less impactful. However, after incorporating our estimation, there is still a slight improvement in forecasting performance. On the other hand, when considering combining two imputation methods with similar original performance, incorporating our method will bring significant gains (as seen in the GEF datasets). It's worth noting that the mean value used as a baseline outperforms other imputation methods in most cases, primarily because the training data input is also based on mean value imputation, facilitating unified and convenient comparisons. In practical applications, we can also apply other advanced imputation methods to the input data and modify the labels based on the estimated benefits.

**II. Comparison between Gain Estimation and Influence Function**. Part I, II, and III Table 2 indicate that discarding a certain number of samples according to the Influence Function can indeed improve the performance of forecasting; however, such improvement is not universal. In the AIR dataset, discarding some data can negatively impact the performance of most methods. This may be due to the small amount of data contained in the AIR dataset, resulting in a greater adverse effect when discarding data. The operation of discarding data can only consider one imputation method, while our method can combine any two imputation methods to achieve better results. Furthermore, the results of repeated experiments show that the strategy of modifying values at specific time steps can make performance more stable, as its variance is significantly smaller than that of discarding data.

In addition to the univariate input results displayed in Table 2, we also include the results of multivariate inputs, which are common in practical applications. For instance, when predicting power loads, temperature is a crucial external variable. A large amount of research has focused on studying the relationship between load and temperature [48, 49]. In this experimental setting, unlike multivariate forecasting, temperature plays an auxiliary role in load forecasting while there is no need to forecast temperature itself. Consequently, we conduct experiments on the GEF dataset, inputting temperature as an external variable into the model to forecast loads. Table 3 presents the performance of our model under multivariate input, which is consistent with the univariate input scenario, and incorporating our method proves to be beneficial.

Table 2: MSE↓ in the downstream forecasting task with univariate input, every experiment is done 3 times.

| Method | Datasets | | | | | |
|---|---|---|---|---|---|---|
| | GEF | ETTH1 | ETTH2 | ELECTRICITY | TRAFFIC | AIR |
| I.Original | | | | | | |
| Mean | 0.1750 | 0.0523 | 0.1797 | 0.1123 | 0.4359 | 0.1508 |
| SAITS | 0.1980(0.0092) | 0.1027(0.0021) | 0.2098(0.0125) | 0.1176(0.0110) | 0.4311(0.0151) | 0.5006(0.0251) |
| BRITS | 0.2021(0.0007) | 0.1692(0.0105) | 0.2384(0.0018) | 0.1503(0.0003) | 0.4535(0.0001) | 0.6979(0.0086) |
| MRNN | 0.2052(0.0001) | 0.2184(0.0016) | 0.2317(0.0001) | - | 0.4540(0.0000) | 0.7965(0.0018) |
| GPVAE | 0.2087(0.0019) | 0.1591(0.0072) | 0.2365(0.0022) | 0.1471(0.0001) | 0.4465(0.0001) | 0.6968(0.0044) |
| USGAN | 0.2048(0.0023) | 0.1549(0.0179) | 0.2238(0.0085) | 0.1447(0.0011) | 0.4742(0.0048) | 0.6840(0.0306) |
| SPIN | 0.2120(0.0029) | 0.2000(0.0509) | 0.2414(0.0327) | 0.1588(0.0113) | 0.4609(0.0148) | 0.6604(0.0802) |
| ImputeFormer | 0.1820(0.0016) | 0.1558(0.0033) | 0.2125(0.0022) | 0.1076(0.0012) | 0.4249(0.0060) | 0.6300(0.0119) |
| II.With Gain estimation | | | | | | |
| Mean+SAITS | **0.1653(0.0008)** | **0.0522(0.0000)** | 0.1797(0.0000) | **0.0957(0.0006)** | **0.4147(0.0023)** | **0.1491(0.0001)** |
| Mean+BRITS | 0.1694(0.0000) | **0.0522(0.0000)** | 0.1795(0.0000) | 0.1068(0.0000) | 0.4318(0.0000) | 0.1507(0.0000) |
| Mean+MRNN | 0.1696(0.0000) | 0.0523(0.0000) | 0.1794(0.0000) | - | 0.4319(0.0000) | 0.1508(0.0000) |
| Mean+GPVAE | 0.1696(0.0000) | **0.0522(0.0000)** | 0.1795(0.0000) | 0.1058(0.0001) | 0.4290(0.0005) | 0.1507(0.0000) |
| Mean+USGAN | 0.1698(0.0001) | **0.0522(0.0000)** | 0.1795(0.0000) | 0.1069(0.0003) | 0.4215(0.0004) | 0.1506(0.0000) |
| Mean+SPIN | 0.1679(0.0016) | 0.0523(0.0001) | **0.1784(0.0000)** | 0.1038(0.0007) | 0.4276(0.0013) | 0.1502(0.0005) |
| Mean+ImputeFormer | 0.1657(0.0003) | **0.0522(0.0000)** | 0.1795(0.0000) | 0.0977(0.0002) | 0.4178(0.0015) | 0.1498(0.0000) |
| III.With Influence Function | | | | | | |
| SATIS+INF | 0.1953(0.0008) | 0.1026(0.0021) | 0.2074(0.0115) | 0.1170(0.0169) | 0.4294(0.0153) | 0.5207(0.0213) |
| BRITS+INF | 0.1952(0.0009) | 0.1637(0.0091) | 0.2326(0.0005) | 0.1302(0.0022) | 0.4419(0.0008) | 0.7110(0.0069) |
| MRNN+INF | 0.1972(0.0002) | 0.1905(0.0017) | 0.2251(0.0003) | - | 0.4431(0.0002) | 0.7758(0.0020) |
| GPVAE+INF | 0.2013(0.0018) | 0.1543(0.0073) | 0.2314(0.0031) | 0.1275(0.0027) | 0.4347(0.0005) | 0.7096(0.0021) |
| USGAN+INF | 0.1984(0.0024) | 0.1486(0.0120) | 0.2191(0.0060) | 0.1263(0.0013) | 0.4597(0.0045) | 0.6961(0.0194) |
| SPIN+INF | 0.2195(0.0046) | 0.2106(0.0422) | 0.2551(0.0428) | 0.1531(0.0224) | 0.4728(0.0194) | 0.7629(0.1007) |
| ImputeFormer+INF | 0.1776(0.0009) | 0.1461(0.0013) | 0.2085(0.0020) | 0.1033(0.0070) | 0.4197(0.0058) | 0.6498(0.0046) |

Table 3: MSE↓ in the downstream forecasting task with multivariate input, every experiment is done 3 times.

| Method | Mean | SAITS | BRITS | MRNN | GPVAE | USGAN | ImputeFormer |
|---|---|---|---|---|---|---|---|
| Original | 0.1845 | 0.1848(0.0022) | 0.2153(0.0136) | 0.2392(0.0321) | 0.2038(0.0006) | 0.1895(0.0004) | 0.1822(0.0012) |
| +Ours | - | 0.1748(0.0002) | 0.1788(0.0000) | 0.1797(0.0005) | 0.1800(0.0000) | 0.1780(0.0004) | **0.1747(0.0011)** |

## 4  Conclusion and Future Work

In this work, we propose to evaluate the imputation values at each time step for the impact on downstream forecasting tasks. On the one hand, our method can accurately estimate the gain of each imputation value without retraining. On the other hand, our method can also combine different time series imputation strategies based on the estimation of gain to obtain better imputation for downstream tasks. To ensure the applicability of this method in practical scenarios, we also provide an accelerated calculation method. In the future, we will focus on further downstream tasks, such as optimization tasks based on prediction values, and build an end-to-end evaluation strategy.

## 5  Acknowledgement

The work was supported in part by the National Key R&D Program of China (2022YFE0141200), in part by the Research Grants Council of the Hong Kong SAR (HKU 27203723), and in part by the Alibaba Group through Alibaba Research Intern Program.

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

# A   Why We Need Task-oriented Imputation

Here we use a toy example to illustrate that, in some cases, we can not directly use the accuracy of imputation instead of downstream tasks to evaluate the imputation method. We want to point out that better imputation accuracy does not always mean better forecasting performance, and we simulate a dataset based on the GEF dataset to illustrate this viewpoint, experimenting with a predicted length of 24. Suppose that we only observed the value at the time step nk (k≥0) and nk+1 (k≥1), just for the convenience of linear interpolation. In the first case(represented by I), we set n = 4, fill the missing value with linear interpolation, and uniformly add Gaussian noise $\mathcal{N}(0.05, 0.3)$. In the second case(represented by II), we set n = 6 and only do the linear interpolation (shown in Figure 2). We put two data sets into MLP and calculated the forecasting error as shown in the following Table 4.

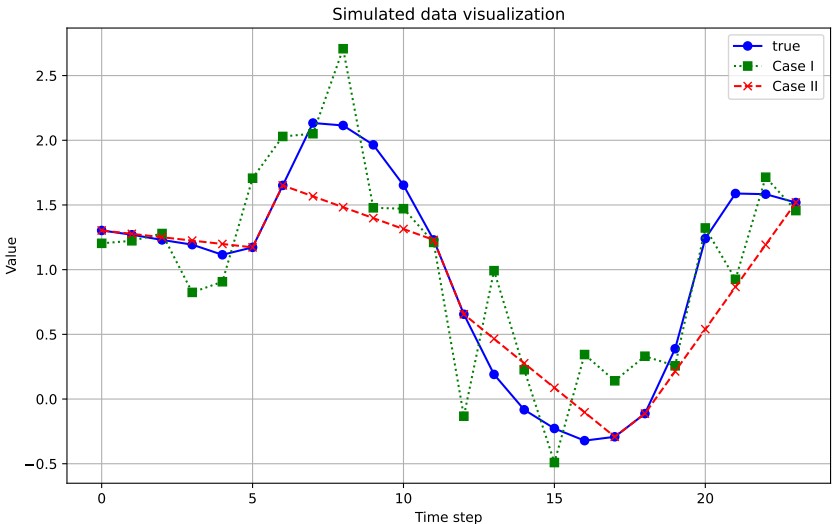

Figure 2: Visualization of simulated data

Table 4: Imputation and Forecasting accuracy on simulated dataset

| MSE↓ | Imputation | Forecasting |
|---|---|---|
| I | 0.1039 | **0.1140** |
| II | **0.0576** | 0.1395 |

# B   Axioms

In [36], there are several axioms desirable for a fair attribution. In this section, we modify them to a version suitable for our task and demonstrate that our method satisfies such properties. Here we use $I(i, l, \boldsymbol{X}_k)$ to represent the impact of the perturbation of the l-th step of the i-th sample on sample $\boldsymbol{X}_k$.

**Definition 1** (Efficiency Axiom). For any model $f(\cdot, \boldsymbol{\theta})$, and test point $\boldsymbol{X}_k^v$, an indication function $I(\cdot, \cdot, \cdot)$ satisfies the efficiency axiom iff:

$$\sum_i^n \sum_l^{L_2} I(i, l) = \sum_{k=1}^m \left( \mathcal{L}\left( f\left(\boldsymbol{X}_k^v, \boldsymbol{\theta}_1\right), \boldsymbol{y}_k^v \right) - \mathcal{L}\left( f\left(\boldsymbol{X}_k^v, \boldsymbol{\theta}_2\right), \boldsymbol{y}_k^v \right) \right)$$

$$\text{s.t. } \boldsymbol{\theta}_1 = \arg\min_{\boldsymbol{\theta}} \sum_{k=1}^n \mathcal{L}\left( f\left(\boldsymbol{X}_k, \boldsymbol{\theta}\right), \boldsymbol{y}_k^{(1)} \right)$$

$$\boldsymbol{\theta}_2 = \arg\min_{\boldsymbol{\theta}} \sum_{k=1}^n \mathcal{L}\left( f\left(\boldsymbol{X}_k, \boldsymbol{\theta}\right), \boldsymbol{y}_k^{(2)} \right)$$

This is a counterpart of the efficiency axioms in Shapley values [50], and our method is naturally satisfying.

**Definition 2** (Self-Explanation Axiom). An indication function $I(\cdot, \cdot, \cdot)$ satisfies the self-explanation axiom iff there exists any training point $\boldsymbol{X}_i$ having no effect on itself, i.e. $I(i, l, \boldsymbol{X}_i) = 0$, the training point should not impact any other points, i.e. $I(i, l, \boldsymbol{X}) = 0$ for all $\boldsymbol{X}$.

Similar to the dummy axiom in the Shapley values, our method naturally satisfies.

**Definition 3** (Symmetric Zero Axiom). An indication function $I(\cdot, \cdot, \cdot)$ satisfies the symmetric zero axiom iff any two training points $\boldsymbol{X}_i, \boldsymbol{X}_j$ such that if $I(i, \cdot, \boldsymbol{X}_i) \neq 0$ and $I(j, \cdot, \boldsymbol{X}_j) \neq 0$, then

$$I(j, \cdot, \boldsymbol{X}_i) = 0 \Longrightarrow I(i, \cdot, \boldsymbol{X}_j) = 0. \tag{10}$$

This situation holds if and only if $K(\boldsymbol{X_i}, \boldsymbol{X_j}) = 0$, therefore, our method satisfies.

**Definition 4** (Symmetric Cycle Axiom). An indication function $I(\cdot, \cdot, \cdot)$ satisfies the symmetric cycle axiom iff for any set of training points $\boldsymbol{X}_{t_1}, \ldots \boldsymbol{X}_{t_k}$, with possible duplicates, and $\boldsymbol{X}_{t_{k+1}} = \boldsymbol{X}_{t_1}$, it holds that:

$$\prod_{i=1}^{k} I\left(t_i, \cdot, \boldsymbol{X}_{t_{i+1}}\right) = \prod_{i=1}^{k} I\left(t_{i+1}, \cdot, \boldsymbol{X}_{t_i}\right) \tag{11}$$

The original definition of $I(\cdot, \cdot, \cdot)$ does not satisfy the above properties, as our goal is to estimate the impact of label perturbations on the loss. However, when removing $\frac{\partial \mathcal{L}(f(\boldsymbol{X}_k, \boldsymbol{\theta}), \boldsymbol{y}_k)}{\partial f(\boldsymbol{X}_k, \boldsymbol{\theta})}$ from the definition, our method satisfies this property.

**Definition 5** (Continuity Axiom). An indication function $I(\cdot, \cdot, \cdot)$ satisfies the continuity axiom iff it is continuous wrt the test data point $\boldsymbol{X}$, for any fixed training point $\boldsymbol{X}_i$ :

$$\lim_{\boldsymbol{X}' \to \boldsymbol{X}} I(j, \cdot, \boldsymbol{X}') = I(j, \cdot, \boldsymbol{X}) \tag{12}$$

**Definition 6** (Irreducibility Axiom). An indication function $I(\cdot, \cdot, \cdot)$ satisfies the irreducibility axiom iff for any number of training points $\boldsymbol{X}_1, \ldots, \boldsymbol{X}_k$,

$$\det \begin{pmatrix} I(1, \cdot, \boldsymbol{X}_1) & I(1, \cdot, \boldsymbol{X}_2) & \ldots & I(1, \cdot, \boldsymbol{X}_k) \\ I(2, \cdot, \boldsymbol{X}_1) & I(2, \cdot, \boldsymbol{X}_2) & \ldots & I(2, \cdot, \boldsymbol{X}_k) \\ \ldots & \ldots & \ldots & \ldots \\ I(k, \cdot, \boldsymbol{X}_1) & I(k, \cdot, \boldsymbol{X}_2) & \ldots & I(k, \cdot, \boldsymbol{X}_k) \end{pmatrix} \geq 0. \tag{13}$$

A sufficient condition for an attribution $A(\cdot)$ to satisfy the irreducibility axiom is for

$$|I(i, \cdot, \boldsymbol{X}_i)| > \sum_{j \neq i} |I(i, \cdot, \boldsymbol{X}_j)| \tag{14}$$

When selecting the NTK kernel, this property naturally satisfies.

## C   Discussion on Remark 1

**Remark 1.** *Given two infinitely differentiable functions $f(\boldsymbol{x})$ and $g(\boldsymbol{x})$ in a bounded domain $D \in R^n$, $||f(\boldsymbol{x}) - g(\boldsymbol{x})||$ is always less than $\epsilon$. For any given $\delta$ and $\epsilon_2$, there exists an $\epsilon$ such that, in the domain $D$, the measure of the region $I$ that satisfying $||\frac{\partial f(\boldsymbol{x})}{\partial \boldsymbol{x}} - \frac{\partial g(\boldsymbol{x})}{\partial \boldsymbol{x}}|| > \delta$ is not greater than $\epsilon_2$, i.e, $m(I) \leq \epsilon_2$.*

**Correctness.** Firstly, we can relax the restrictions on the function by requiring that each dimension of the function on $R^n$ be continuous. Then, we can simplify the problem into a one-dimensional case on $R$. Secondly, we can let $h(x) = f(x) - g(x)$ (note that $|h(x)| < \epsilon$), and then our problem can be transformed into proving that for any given $\delta$ and $\epsilon_2$, there exists an $\epsilon$ such that, in the interval $[a, b]$, the measure of the region satisfying $|h'(x)| > \delta$ is less than $\epsilon_2$.

Let the domain $I$ represent the set of all $x$ that satisfy $|h'(x)| > \delta$. We first need to prove that $I$ can be rewritten as the union of several disjoint intervals $I_i$ that satisfy

$$\forall I_i, \forall x \in I_i, h(x) > \delta \quad or \quad h(x) < -\delta.$$

Since $h'(x)$ itself is a continuous function, the division is obvious. The problem is that we need to prove the number of $I_i$ is countable.

Assuming $I_i$ is uncountable, we perform n bisection on $[a, b]$ to obtain numbers of smaller intervals like $[a + (\frac{1}{2})^n, a + (\frac{1}{2})^{n+1}]$. If $I_i$ is not countable, then no matter how large $n$ is, we can always find two points $x$ and $y$ in the interval that satisfy $h'(x) > \delta, h'(y) > \delta$ (here we mainly consider greater cases, the smaller case is the same), then

$$|x - y|\delta \leq |h(x) - h(y)| \leq |h(x)| + |h(y)| \leq 2\epsilon,$$

and this equation does not hold as $\epsilon \to 0$, which makes contradiction. Therefore, we can rewrite $I$ as $\bigcup_{i=1}^{\infty} I_i$. To simplify, assume that $I$ can be written in the form of a finite number of unions $I = \bigcup_{i=1}^{N} I_i$. If the remark does not hold, we have $\max_i m(I_i) = k$, then $Nk \geq \epsilon_2$. Obviously, $|\delta k| \leq 2\epsilon$, then $k \leq |\frac{2\epsilon}{\delta}|$. Finally, we have

$$N|\frac{2\epsilon}{\delta}| \geq Nk \geq \epsilon_2,$$

which makes contradiction as $\epsilon \to 0$. If the number of $I_i$ is infinite, we can always find $N$ that is big enough to satisfy $Nk \geq \epsilon_2$ and the rest is the same.

**Discussion on Application.** As shown in the analysis above, for some given $\delta$ and $\epsilon_2$, we only require that $|\epsilon| < |x - y|\delta$ and $|\epsilon| < |\frac{\epsilon_2 \delta}{2N}|$. Even though the real boundary is highly related to the specific scenarios and is difficult to tell in real applications, we can have a look at some widely used examples. Here, we mainly discuss two kinds of optimizers that are widely used in neural network training and they are SGD [51] and Adam [52].

For the loss function $\mathcal{L}$ to be set as MSE. Note that we use $X$ and $X_{test}$ to represent the input data in the training set and test set, separately. $y$ is the training label and $y_{i,l}$ is the value of a time step in it. We use t to represent the training epoch and there will be T epochs total. Our goal is to use $g(X_{test}, y_{i,l})$ to approximate $f(X_{test}, \theta_T, y_{i,l})$ first and then use $\frac{\partial g(X_{test}, y_{i,l})}{\partial y_{i,l}}$ to approximate $\frac{\partial f(X_{test}, \theta_T, y_{i,l})}{\partial y_{i,l}}$.

During the training process of a forecasting model, we let $h_t(y_{i,l})$ represent $\frac{\partial \mathcal{L}(f(X, \theta_t, y_{i,l}), y)}{\partial \theta_t}$ and $\frac{\partial^2 h_t(y_{i,l})}{\partial y_{i,l}^2}$ will be zero. Recalling that our goal is to approximate the gradient $\frac{\partial f(X_{test}, \theta_T, y_{i,l})}{\partial y_{i,l}}$ by approximating $f(X_{test}, \theta_T, y_{i,l})$ first and then take the gradient. However, the function that we really want to approximate is $\frac{\partial \theta_T}{\partial y_{i,l}}$ since $\frac{\partial f(X_{test}, \theta_T, y_{i,l})}{\partial y_{i,l}} = \frac{\partial f(X_{test}, \theta_T, y_{i,l})}{\partial \theta_T} \frac{\partial \theta_T}{\partial y_{i,l}}$ and $\frac{\partial f(X_{test}, \theta_T, y_{i,l})}{\partial \theta_T}$ is a constant for given $X_{test}$.

For SGD (as well as its variants SGD with momentum), $\theta_T = \theta_0 - \sum_{t=1}^{T} \eta h_t(y_{i,l})$. Therefore, the $\frac{\partial^2 \theta_T}{\partial y_{i,l}^2} = 0$, which means that the gradient of the function we want to approximate is constant. In this case, our approximation will be pretty good.

For Adam, on the one hand, [35] has claimed that the first-order approximation in the SGD situation remains valid, as long as a small step-size $\eta$ is used in the update. On the other hand, let $\theta_T = \theta_0 - \sum_{t=1}^{T} \eta v_t(h_t(y_{i,l}))$, where $v_t$ represent the terms in Adam. In this situation, $\frac{\partial \theta_T}{\partial y_{i,l}}$ will be an algebraic function with only a finite number of monotonic intervals. Therefore, for any given $\delta$ and $\epsilon$, the $\epsilon_2$ will not be really high since the $y_{i,l}$ that makes $||\frac{\partial f(X_{test}, \theta_T, y_{i,l})}{\partial y_{i,l}} - \frac{\partial g(X_{test}, y_{i,l})}{\partial y_{i,l}}|| > \delta$ can only appear near the local extreme point, whose number is finite.

# D    Implementation Details

Table 5 summarizes the dataset partitioning we used. Except for the GEF data, the rest are multivariate datasets. We forecast the 'OT' sequences in ETH1 and ETH2, as well as the combined electricity

consumption of all users in ELECTRITY, the average occupancy rate of all roads in TRAFFIC, and the temperature series in the AIR dataset.

## D.1 Datasets for forecasting

Table 5: Datasets used in the forecasting task.

| Dataset | Training | Validation | Test |
|---|---|---|---|
| GEF | 2013-01-01 01:00~2014-01-01 00:00 | 2014-01-01 00:00~2014-09-01 01:00 | 2014-09-01 01:00~2015-01-01 00:00 |
| ETTH1 | 2016-07-01 00:00~2017-07-01 00:00 | 2017-07-01 00:00~2018-03-26 01:00 | 2018-03-26 01:00~2018-06-26 19:00 |
| ETTH2 | 2016-07-01 00:00~2017-07-01 00:00 | 2017-07-01 00:00~2018-03-26 01:00 | 2018-03-26 01:00~2018-06-26 19:00 |
| ELECTRICITY | 2012-01-01 00:00~2013-08-01 00:00 | 2013-08-01 00:00~2014-06-26 01:00 | 2014-06-26 01:00~2014-12-31 23:00 |
| TRAFFIC | 2016-07-01 02:00~2017-12-31 23:00 | 2017-12-31 23:00~2018-04-01 01:00 | 2018-04-01 01:00~2018-07-02 01:00 |
| AIR | 2004-03-10 18:00~2004-12-01 00:00 | 2004-12-01 00:00~2005-02-01 00:00 | 2005-02-01 00:00~2005-04-04 13:00 |

## D.2 Implementation of time series forecasting model

We include two models in our experiment. The first one is a 3-layer MLP in which the input size and output size are both 24 while the hidden size is 128. In addition, we mainly apply the simple and high-performance DLiner with default setting in [47] as our forecasting model backbone. In addition, to adapt to situations where the input and output dimensions are different, we constructed an output layer at the end of the DLiner model, mapping the output of multiple variables to the output of a specified dimension. We use the torch.SGD optimizer [53] to optimize the parameters of the model, where the learning rate is set to 0.1. The maximum epochs for each training are 300, and the patience is set to 10.

## D.3 Implementation of time series imputation model

We have introduced a total of five imputation methods for comparison, and all experiments were based on pyPOTS [45] except for SPIN. The hyperparameters for each method are set as shown in Table 6. In addition, referring to [13], we set the total training epoch to 100 and the patience to 10, while other hyperparameters are the default setting.

Table 6: Hyperparameters of the time series imputation.

| Method | Hyperparameters |
|---|---|
| SAITS | n_layers=2, d_model=64, d_ffn=32, n_heads=4, d_k=16, d_v=16, dropout=0.1 |
| BRITS | rnn_hidden_size=64 |
| MRNN | rnn_hidden_size=64 |
| GPVAE | latent_size=64 |
| USGAN | rnn_hidden_size=64 |
| SPIN | hidden_size=64 |
| ImputeFormer | n_layers = 2,d_input_embed =64,d_learnable_embed = 64 ,d_proj = 32,d_ffn = 64 ,n_temporal_heads = 4 |

## D.4 Implementation of our estimation

In section 2.2, we gives the estimation of $I(i, l)$ as follows,

$$-\frac{1}{n} \frac{\partial^2 \mathcal{L}\left(f\left(\boldsymbol{X}_i, \boldsymbol{\theta}\right), \boldsymbol{y_i}\right)^T}{\partial f\left(\boldsymbol{X}_i, \boldsymbol{\theta}_t\right) \partial y_{i,l}} \frac{\partial \mathcal{L}(f(\boldsymbol{X}_k^v, \boldsymbol{\theta}), \boldsymbol{y}_k^v)}{\partial f(\boldsymbol{X}_k^v, \boldsymbol{\theta})} \underbrace{\frac{\partial f\left(\boldsymbol{X}_i, \boldsymbol{\theta}\right)}{\partial \boldsymbol{\theta}} \frac{\partial f(\boldsymbol{X}_k^v, \boldsymbol{\theta})}{\partial \boldsymbol{\theta}}^T}_{NTK kernel}. \quad (15)$$

However, depending on the solution to the optimization problem (7), we may have different forms of estimation for $I(i, l)$. Referring to [36, 35], when we no longer only consider the downstream model parameters $\boldsymbol{\theta}$ at the moment of training convergence but also consider the entire training process, we can obtain another form of solution to the optimization problem that

$\hat{\alpha_{i,l}} = -\sum_{t\in[T]:i\in B^{(t)}} \frac{\eta^{(t)}}{|B^{(t)}|} \frac{\partial^2 \mathcal{L}\left(f^{(t-1)}(\boldsymbol{X}_i,\boldsymbol{\theta}),\boldsymbol{y_i}\right)}{\partial f^{(t-1)}(\boldsymbol{X}_i,\boldsymbol{\theta})\partial y_{i,l}}$, where $t$, $B^{(t)}$, and $\eta^{(t)}$ here represent the $t$ th epoch, the batch size, and the learning rate, respectively. And the $I(i,l)$ in this situation will be

$$
\begin{aligned}
&- \sum_{t\in[T]:i\in B^{(t)}} \frac{\eta^{(t)}}{\left|B^{(t)}\right|} \frac{\partial^2 \mathcal{L}\left(f^{(t-1)}\left(\boldsymbol{X}_i,\boldsymbol{\theta}\right),\boldsymbol{y_i}\right)^T}{\partial f^{(t-1)}\left(\boldsymbol{X}_i,\boldsymbol{\theta}_t\right)\partial y_{i,l}} \frac{\partial \mathcal{L}(f(\boldsymbol{X}_k^v,\boldsymbol{\theta}),\boldsymbol{y}_k^v)}{\partial f(\boldsymbol{X}_k^v,\boldsymbol{\theta})} \frac{\partial f^{(t-1)}\left(\boldsymbol{X}_i,\boldsymbol{\theta}\right)^\top}{\partial\boldsymbol{\theta}} \frac{\partial f(\boldsymbol{X}_k^v,\boldsymbol{\theta})}{\partial\boldsymbol{\theta}}\Bigg|_{\boldsymbol{\theta}=\boldsymbol{\theta}^{(t)}}, \\
=&- \sum_{t\in[T]:i\in B^{(t)}} \frac{\eta^{(t)}}{\left|B^{(t)}\right|} \frac{\partial^2 \mathcal{L}\left(f^{(t-1)}\left(\boldsymbol{X}_i,\boldsymbol{\theta}\right),\boldsymbol{y_i}\right)}{\partial f^{(t-1)}\left(\boldsymbol{X}_i,\boldsymbol{\theta}_t\right)\partial y_{i,l}} \frac{\partial f^{(t-1)}\left(\boldsymbol{X}_i,\boldsymbol{\theta}\right)^\top}{\partial\boldsymbol{\theta}} \frac{\partial \mathcal{L}(f(\boldsymbol{X}_k^v,\boldsymbol{\theta}),\boldsymbol{y}_k^v)}{\partial\boldsymbol{\theta}}\Bigg|_{\boldsymbol{\theta}=\boldsymbol{\theta}^{(t)}}.
\end{aligned}
\tag{16}
$$

Compared to the original estimation, repeated calculations will bring a significant computational burden. However, based on our acceleration method, the time required for such multiple calculations is still controlled within a reasonable range. In application, we calculate for each parameter updates in each epoch.

### D.5    Implementation of Influence Function

In section 3.3.1, we compared the performance of our method with the Influence Function that we modified. Below, we will describe how to modify the Influence Function to fit our task.

For a training point $(\boldsymbol{X},\boldsymbol{y})$, define $\boldsymbol{y}_{l,\delta} \overset{\text{def}}{=} [y_1,\cdots,y_l+\delta,\cdots,y_{L_2}]$. Consider the perturbation $\boldsymbol{y} \mapsto \boldsymbol{y}_{l,\delta}$ and let $\boldsymbol{\theta}_{\epsilon,\delta} \overset{\text{def}}{=} \arg\min_{\theta\in\Theta} \frac{1}{n}\sum_{i=1}^n L\left(f(\boldsymbol{X}_i,\boldsymbol{\theta}),\boldsymbol{y}_i\right) + \epsilon L\left(f(\boldsymbol{X},\boldsymbol{\theta}),\boldsymbol{y}_{l,\delta}\right) - \epsilon L\left(f(\boldsymbol{X},\boldsymbol{\theta}),\boldsymbol{y}_{l,\delta}\right)$. Then we have

$$
\frac{d\hat{\boldsymbol{\theta}}_{\epsilon,\delta}}{d\epsilon}\Bigg|_{\epsilon=0} = -H_{\boldsymbol{\theta}}^{-1}\left(\nabla_{\boldsymbol{\theta}}L(f(\boldsymbol{X},\boldsymbol{\theta}),\boldsymbol{y})\right) - \nabla_{\boldsymbol{\theta}}L(f(\boldsymbol{X},\boldsymbol{\theta}),\boldsymbol{y})) \tag{17}
$$

$$
\approx -H_{\boldsymbol{\theta}}^{-1}\left[\nabla_{y_l}\nabla_{\boldsymbol{\theta}}L(f(\boldsymbol{X},\boldsymbol{\theta}),\boldsymbol{y})\right]\delta. \tag{18}
$$

Therefore, $I(l,\boldsymbol{X}_{test}) = -\nabla_{\boldsymbol{\theta}}L\left(f(\boldsymbol{X}_{test},\boldsymbol{\theta}),\boldsymbol{y}_{test}\right)^\top H_{\hat{\boldsymbol{\theta}}}^{-1}\left[\nabla_{y_l}\nabla_{\boldsymbol{\theta}}L(f(\boldsymbol{X},\hat{\boldsymbol{\theta}}),\boldsymbol{y})\right]\delta$. Note that we applied the Conjugate gradients mentioned in [34] to accelerate its computation and compare it with our methods.

### D.6    Hareware usage

We use 1 NVIDIA GTX 4090 GPU with 24GB of memory for all our experiments.

## E    Potential Social Impact

Our estimation may not be $100\%$ accurate compared to the actual situation, so it is possible to introduce bias in the evaluation among different imputation strategies, which may further have adverse effects on downstream tasks.

## F    Supplementary Experimental Results

### F.1    Acceleration method

#### F.1.1    Performance of the acceleration method

In our practice, we mainly examine the benefits of modifying each time step on downstream tasks. Therefore, we mainly focus on whether the gain estimation is positive or negative without providing precise values. Based on this idea, we provide methods for accelerating calculations in Section 2.3. Here, we present a comparison between the accelerated estimate and the original estimate. Note that we conduct this experiment on three datasets and they are GEF, ELECTRICITY, and a generated time series, denoted by Brown, based on the following Python code.

```
from fbm import FBM
f = FBM(n=2281, hurst=0.75, length=1, method='daviesharte')
```

The result is shown in the Figure 3. Note that here we also use correlation and accuracy mentioned in the main text.

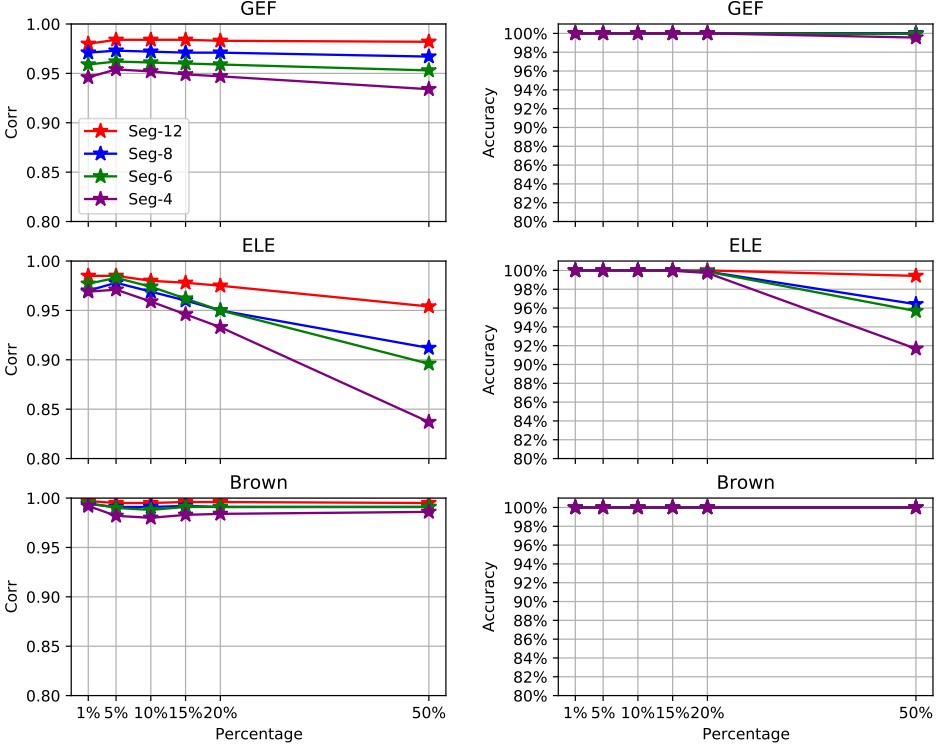

Figure 3: The correlation and accuracy comparison between the estimation of our original method and the acceleration method.

### F.1.2 MSE comparison on downstream forecasting task

Table 7: MSE↓ in the downstream forecasting task.

| Method | Datasets | | | | | |
| --- | --- | --- | --- | --- | --- | --- |
| | GEF | ETTH1 | ETTH2 | ELECTRICITY | TRAFFIC | AIR |
| I.Original | | | | | | |
| Mean | 0.1750 | 0.0523 | 0.1797 | 0.1123 | 0.4359 | 0.1508 |
| SAITS | 0.1980(0.0092) | 0.1027(0.0021) | 0.2098(0.0125) | 0.1176(0.0110) | 0.4311(0.0151) | 0.5006(0.0251) |
| BRITS | 0.2021(0.0007) | 0.1692(0.0105) | 0.2384(0.0018) | 0.1503(0.0003) | 0.4535(0.0001) | 0.6979(0.0086) |
| MRNN | 0.2052(0.0001) | 0.2184(0.0016) | 0.2317(0.0001) | - | 0.4540(0.0000) | 0.7965(0.0018) |
| GPVAE | 0.2087(0.0019) | 0.1591(0.0072) | 0.2365(0.0022) | 0.1471(0.0001) | 0.4465(0.0001) | 0.6968(0.0044) |
| USGAN | 0.2048(0.0023) | 0.1549(0.0179) | 0.2238(0.0085) | 0.1447(0.0011) | 0.4742(0.0048) | 0.6840(0.0306) |
| SPIN | 0.2120(0.0029) | 0.2000(0.0509) | 0.2414(0.0327) | 0.1588(0.0113) | 0.4609(0.0148) | 0.6604(0.0802) |
| ImputeFormer | 0.1820(0.0016) | 0.1558(0.0033) | 0.2125(0.0022) | 0.1076(0.0012) | 0.4249(0.0060) | 0.6300(0.0119) |
| II.With Seg-4 Gain estimation | | | | | | |
| Mean+SAITS | **0.1666(0.0007)** | **0.0522(0.0000)** | 0.1796(0.0000) | **0.0972(0.0006)** | **0.4182(0.0022)** | **0.1490(0.0001)** |
| Mean+BRITS | 0.1704(0.0000) | **0.0522(0.0000)** | 0.1795(0.0000) | 0.1078(0.0000) | 0.4332(0.0000) | 0.1507(0.0000) |
| Mean+MRNN | 0.1707(0.0000) | 0.0523(0.0000) | 0.1795(0.0000) | - | 0.4333(0.0000) | 0.1508(0.0000) |
| Mean+GPVAE | 0.1708(0.0000) | **0.0522(0.0000)** | 0.1795(0.0000) | 0.1069(0.0001) | 0.4308(0.0004) | 0.1507(0.0000) |
| Mean+USGAN | 0.1704(0.0001) | **0.0522(0.0000)** | 0.1795(0.0000) | 0.1076(0.0002) | 0.4251(0.0002) | 0.1506(0.0000) |
| Mean+SPIN | 0.1693(0.0013) | 0.0523(0.0001) | 0.1800(0.0003) | 0.1047(0.0001) | 0.4302(0.0010) | 0.1502(0.0005) |
| Mean+ImputeFormer | **0.1666(0.0002)** | **0.0522(0.0000)** | **0.1794(0.0000)** | 0.0991(0.0002) | 0.4203(0.0014) | 0.1498(0.0000) |
| III.With Seg-2 Gain estimation | | | | | | |
| Mean+SAITS | 0.1686(0.0005) | **0.0522(0.0000)** | 0.1799(0.0001) | 0.1003(0.0005) | 0.4212(0.0017) | 0.1491(0.0001) |
| Mean+BRITS | 0.1724(0.0000) | **0.0522(0.0000)** | 0.1795(0.0000) | 0.1105(0.0000) | 0.4355(0.0000) | 0.1507(0.0000) |
| Mean+MRNN | 0.1730(0.0000) | 0.0523(0.0000) | 0.1795(0.0000) | - | 0.4356(0.0000) | 0.1508(0.0000) |
| Mean+GPVAE | 0.1730(0.0000) | **0.0522(0.0000)** | 0.1795(0.0000) | 0.1097(0.0001) | 0.4335(0.0003) | 0.1507(0.0000) |
| Mean+USGAN | 0.1724(0.0001) | **0.0522(0.0000)** | 0.1795(0.0000) | 0.1098(0.0003) | 0.4290(0.0001) | 0.1506(0.0000) |
| Mean+SPIN | 0.1733(0.0008) | 0.0523(0.0000) | 0.1803(0.0004) | 0.1093(0.0003) | 0.4343(0.0001) | 0.1503(0.0005) |
| Mean+ImputeFormer | 0.1688(0.0004) | 0.0523(0.0001) | 0.1795(0.0000) | 0.1021(0.0002) | 0.4231(0.0008) | 0.1498(0.0000) |

Table 7 summarizes the performance of our acceleration methods Seg-4 and Seg-2 in downstream forecasting tasks. Overall, performance gradually improves as the number of segmented segments

gradually increases. In addition, even if it is only divided into two segments, our method can still bring some gain with minimal additional computational burden.

### F.1.3 A larger dataset

For large-scale data, we applied our method to a 15-minute resolution UCI electricity dataset (with approximately 100000 training points) and we adjusted our experimental setup to input 96 points and output 96 points, and here is the result.

Table 8: MSE ↓ comparison on larger dataset.

| ELETRICITY | Mean | SAITS | +IF | +ours | +ours_seg_16 | +ours_seg_8 |
|---|---|---|---|---|---|---|
| MSE↓ | 0.249 | 0.307 | 0.248 | 0.238 | **0.236** | 0.237 |
| Time(s) | - | - | 126.33 | 1053.28 | 264.27 | **118.53** |

### F.2 Additional missing rate

In addition to the $40\%$ missing rate in the main experiment, we also conduct several experiments in the ELECTRICITY dataset with missing rates in $[30\%, 50\%, 60\%]$.

Table 9: MSE ↓ comparison on different missing rate.

| ELETRICITY | STRATEGY | Mean | SAITS | BRITS | GPVAE | USGAN | ImputeFormer |
|---|---|---|---|---|---|---|---|
| 30% | - | 0.0878 | 0.0988(0.0107) | 0.1083(0.0016) | 0.1077(0.0002) | 0.1045(0.0006) | 0.0863(0.0005) |
|  | +ours | - | **0.0798(0.0008)** | 0.0855(0.0000) | 0.0851(0.0002) | 0.0853(0.0000) | 0.0814(0.0001) |
| 50% | - | 0.1410 | 0.1686(0.0320) | 0.2082(0.0010) | 0.2044(0.0021) | 0.2071(0.0099) | 0.1347(0.0010) |
|  | +ours | - | **0.1126(0.0026)** | 0.1313(0.0000) | 0.1289(0.0005) | 0.1319(0.0009) | 0.1152(0.0002) |
| 60% | - | 0.1938 | 0.2441(0.0350) | 0.3094(0.0005) | 0.3057(0.0056) | 0.3032(0.0168) | 0.1889(0.0011) |
|  | +ours | - | **0.1351(0.0020)** | 0.1724(0.0001) | 0.1691(0.0015) | 0.1741(0.0016) | 0.1431(0.0011) |

### F.3 Combination with robust time series forecasting

In addition to our solution, some kinds of methods, such as robust time series forecasting, to deal with missing (anomaly) values have been proposed these days. Here we combine our method with [16], which is one of the SOTA of such kind of method to illustrate that this kind of method is not contradictory to our approach but can be combined. Note that the hyperparameters are the same as the original paper in [16] and we replace the dataset with ours.

Table 10: MSE ↓ comparison on the ELECTRICITY dataset combining our method and RobustTSF.

| MSE↓ | Mean | SAITS | BRITS | USGAN | GPVAE | SPIN | ImputeFormer |
|---|---|---|---|---|---|---|---|
| +RobustTSF | 0.056 | 0.050 | 0.092 | 0.084 | 0.099 | 0.053 | 0.076 |
| +RobustTSF+ours | - | **0.046** | 0.050 | 0.052 | **0.046** | 0.048 | 0.051 |

### F.4 Illustration on multivariate forecasting

We conduct our method on multivariate forecasting tasks and give a small example of the ELECTRICITY dataset. Note that we apply our method to the first three users (columns) in the dataset.

Table 11: MSE ↓ comparison on multivariate forecasting.

| MSE↓ | Mean | SAITS | BRITS | GPVAE | USGAN | ImputeFormer |
|---|---|---|---|---|---|---|
| Original | 0.1776 | 0.2252 | 0.3404 | 0.3404 | 0.3355 | 0.2231 |
| +Ours | **-** | 0.1497 | 0.1704 | 0.1717 | 0.1704 | **0.1470** |

### F.5 Forecasting results

Figures 4, 5, 6, 7, 8, and 9, show the visualization of the forecasting performance of the model. We mainly compared the combination of the SAITS method and the baseline model with the original SATIS method, and it can be seen that combining the two imputations will bring benefits to the forecasting.

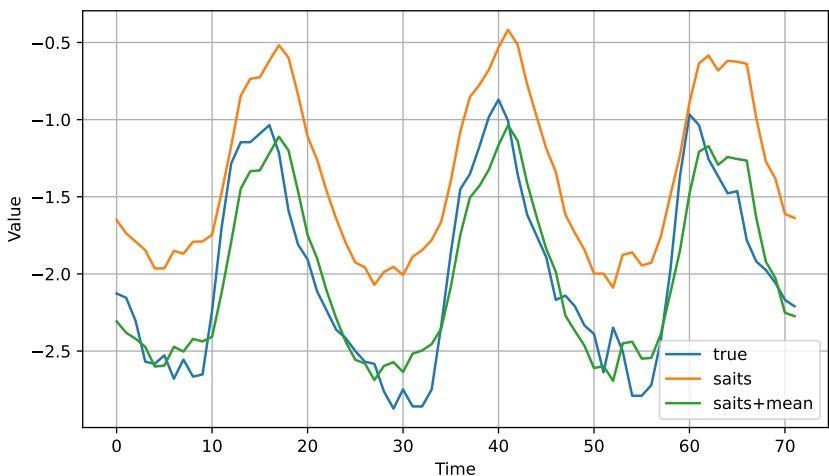

Figure 4: Visualization of forecasting result on AIR

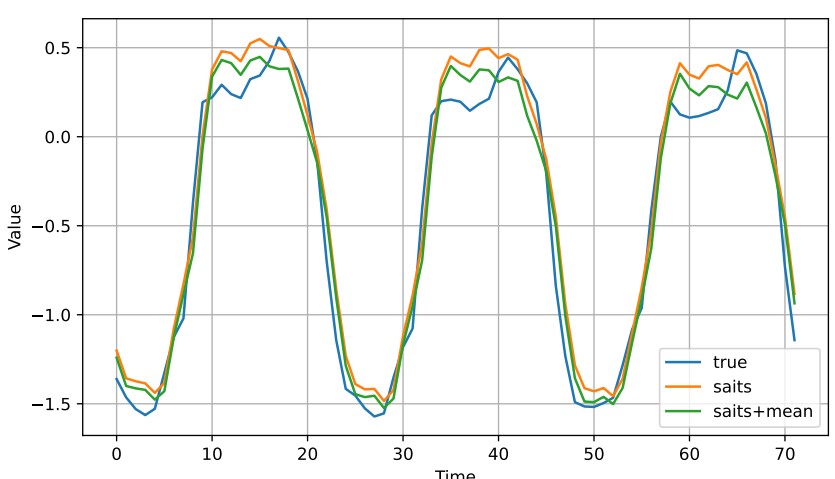

Figure 5: Visualization of forecasting result on ELECTRICITY

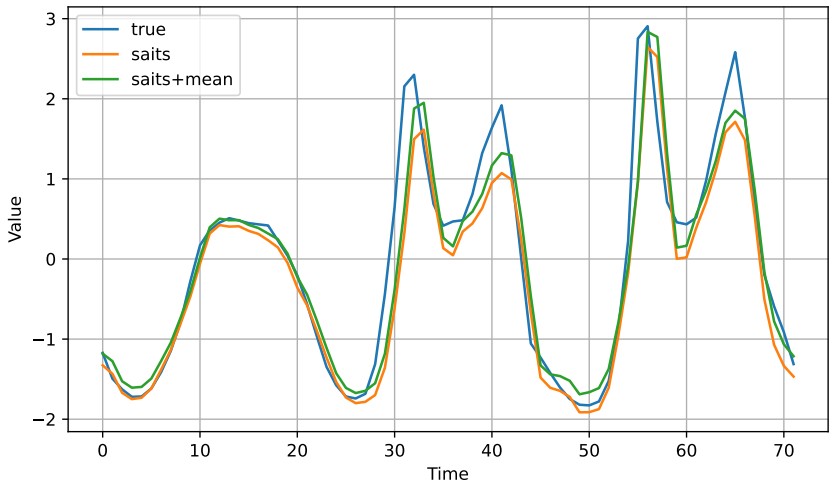

Figure 6: Visualization of forecasting result on Traffic

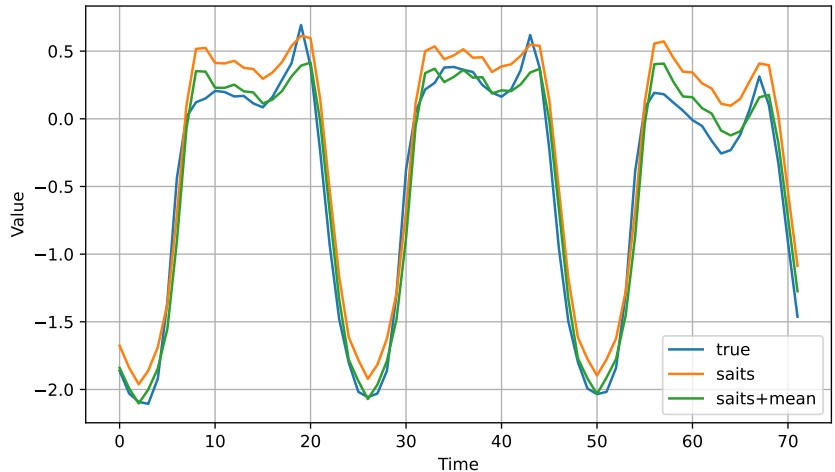

Figure 7: Visualization of forecasting result on GEF

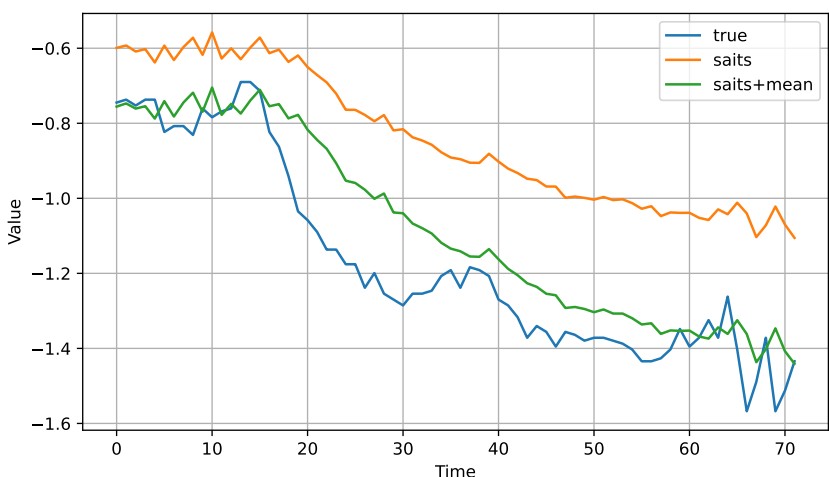

Figure 8: Visualization of forecasting result on ETTH1

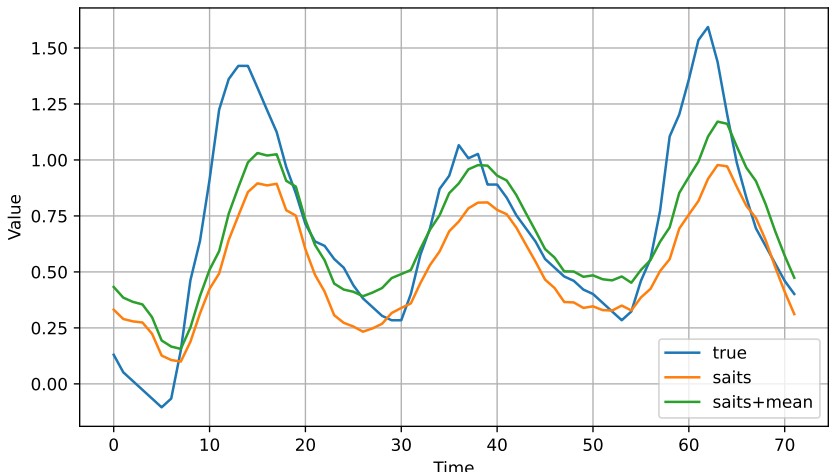

Figure 9: Visualization of forecasting result on ETTH2

