# OpenReview forum: "Task-oriented Time Series Imputation Evaluation via Generalized Representers"
_NeurIPS.cc/2024/Conference — NeurIPS 2024 poster_

### Official Review · Reviewer_xbnY · 2024-06-19

**Soundness:** 3
**Presentation:** 3
**Contribution:** 3
**Rating:** 5
**Confidence:** 4

**Summary:**

This paper studies the problem of evaluating time series imputation methods in terms of the performance on downstream tasks. It proposes a fine-grained metric and uses RKHS to efficiently estimate the metric. Experiments demonstrate that the proposed method achieves better estimation than the influence function approach.

**Strengths:**

S1: This paper studies an interesting perspective for the evaluation of imputation methods: performance on downstream tasks.

S2: This paper borrows an RKHS-based method from the field of model explanation. This may be of independent interest even for a broader research community.

S3: The authors have provided documented code to facilitate reproducibility.

**Weaknesses:**

W1: The proposed Section 2 & Appendix A are basically Sections 4.1 & 3 in [28], respectively. Thus, the paper should clearly attribute these contributions to [28].
- [28] Tsai et al. Sample based explanations via generalized representers. NeurIPS, 2023.

W2: Since this paper focuses on evaluation, it should provide a comprehensive evaluation. However, this paper mainly uses older imputation methods, and a number of latest imputation methods are missing (e.g., [SPIN,PoGeVon]).
- [SPIN] Marisca et al. Learning to reconstruct missing data from spatiotemporal graphs with sparse observations. NeurIPS, 2022.
- [PoGeVon] Wang et al. Networked time series imputation via position-aware graph enhanced variational autoencoders. KDD, 2023.

Minor issue: Some sentences are unclear and hard to understand. For example, Lines 103-104 wrote "$X_i^{D\times L_1}$ corresponds to ... $L_1$ temporal intervals," but what they actually mean seems to be "one time range of length $L_1$." It might be better if the authors further polish their writing to improve clarity.

**Questions:**

See weaknesses and limitations.

**Limitations:**

L1: The proposed method uses RKHS to approximate the function. However, it seems difficult to check in practice whether the approximation is accurate or not.

L2: The problem setting does not seem to be well motivated. As what people usually care about is just an aggregated metric (like Table 2), it is unclear why it is worthy to evaluate in such a fine granularity (i.e., I(i,l)) using more computation.

L3: This paper focuses on missing labels but does not consider missing input data. As the authors have dicussed in Section 1, missing input data can have a big impact on forecasting results if the time of missing data is close to the test data.

---

> ### Author Rebuttal · Authors · 2024-08-06
>
> We thank the reviewer for the valuable feedback and we have addressed each of the concerns raised by the reviewer as outlined below.
>
> >Weakness 1
>
> Thank you for your comments and we will revise the relevant wording in the revised version to attribute more properly. In addition, we would like to clarify the differences between us and [1], which can be summarized into three points.
>
> **New research objects and techniques**: [1] mainly focus on how to examine the impact of each sample on test results, while we focus on the impact of sample labels on test results and provide **Lemma 1** to ensure the feasibility of our approach.
>
> **New application scenarios**: [1] mainly focus on the influence of samples. When the samples show negative effects, the only way is to discard the negative samples, which may also discard some useful information. We focus on time series imputation tasks, obtaining imputation that is more advantageous at each time step, and minimizing information loss to the extent possible.
>
> **New perspective**: [1] mainly focuses on how to interpret the output of neural networks, and we propose the idea of using downstream forecasting tasks to evaluate time series imputation methods, which is also a current focus in this field [2].
>
> >Weakness 2
>
> We have added two new time series imputation methods as supplements to Table 2(also shown in the pdf uploaded), including the Spin and the ImputeFormer(IF) [3] published in KDD 2024. As for PoGeVon, we are unable to implement it in a short time due to the lack of relevant open-source code. If the reviewer thinks it's necessary, we can try adding it to the revised version of the paper.
>
> |            |     GEF    |    ETTH1   |    ETTH2   |     ELE    |   TRAFFIC  |     AIR    |
> |:----------:|:----------:|:----------:|:----------:|:----------:|:----------:|:----------:|
> |    Mean    |   0.1750   |   0.0523   |   0.1797   |   0.1123   |   0.4359   |   0.1508   |
> |    SAITS   |   0.1980   |   0.1027   |   0.2098   |   0.1176   |   0.4311   |   0.5006   |
> |    SPIN    |   0.2120   |   0.2000   |   0.2414   |   0.1588   |   0.4609   |   0.6604   |
> |     IF     |   0.1820   |   0.1558   |   0.2125   |   0.1076   |   0.4249   |   0.6300   |
> | SAITS+ours | **0.1653** | **0.0522** |   0.1797   | **0.0957** | **0.4147** | **0.1491** |
> |  SPIN+ours |   0.1679   |   0.0523   | **0.1784** |   0.1038   |   0.4276   |   0.1504   |
> |   IF+ours  |   0.1657   | **0.0522** |   0.1795   |   0.0977   |   0.4178   |   0.1498   |
>
> From the experimental results, it can be seen that the conclusion is consistent with the original paper, and our method can help the imputation method achieve better results.
>
> >Limitations 1
>
> We can analyze the approximation error from two parts.
>
> 1. In Section 2, we approximate label $y_{i,l}$ using a first-order approximation. This is reasonable for the MSE loss function, as the second-order term doesn't affect model parameters($\frac {\partial^2\frac{\partial \mathcal{L}(f_\theta(x),y)}{\partial \theta }}{\partial y_{i,l}^2} = 0$). Lemma 1 shows that with small fitting errors, we can make the error between derivatives of fitted and original functions small with any probability. In practice, we only need to determine if imputation at each step is beneficial, focusing on positive or negative gains. This implies that our estimation can bring significant gains to downstream tasks.
>
> 2. The above analysis indicates that our fitting error depends on the error of using the NTK kernel function to fit the neural network. It is difficult to analyze this error. However, many related works have demonstrated the rationality of doing so through extensive experiments[1].
>
> >Limitations 2
>
> We admit that aggregated metric has many advantages like simplicity. However, detailed information is also important in the real world. For example, in the energy system, the peak load(which is a small part of all the load series) caused by extreme weather these days has received increasing attention, as it is the scenario most likely to lead to the collapse of the energy system and result in significant economic losses[4]. Similarly, in the field of healthcare, fine-grained information also helps doctors make correct judgments, which is closely related to patients' lives. All of these indicate the necessity of conducting a detailed analysis of time series.
>
> >Limitations 3
>
> As the reviewer said, the impact of $x_{j,k}$ in the input on the prediction is influenced by its location, and in practical applications, we have no way to control the position where missing values appear in the input, which is not conducive to evaluating different imputation methods. Therefore, our main focus here is on the impact of label y. However, our method can also be easily applied to examine the impact of input value $x_{j,k}$, simply replacing $y_{i,l}$ in $\frac{\partial^2 \mathcal{L}\left(f\left(X_i, \theta\right), y_{i}\right)^T}{\partial f\left(X_i, \theta \right) \partial y_{i, l}}$ (Eq.(8) in the original paper) with $x_{j,k}$ will be applicable. Here we give an example of how our approach can help improve more than 20% of the accuracy in the table below.
>
> | Replacing input x |   ELE  |
> |:-----------------:|:------:|
> |       SAITS       | 0.1492 |
> |     SAITS+ours    | **0.1120** |
>
> >Minor issue
>
> Thank you very much and we will polish our paper in the revised version.
>
> [1] Tsai C P, Yeh C K, Ravikumar P. Sample-based explanations via generalized representers[J]. Advances in Neural Information Processing Systems, 2024, 36.
>
> [2] Wang J, Du W, Cao W, et al. Deep learning for multivariate time series imputation: A survey[J]. arXiv preprint arXiv:2402.04059, 2024.
>
> [3] Nie, T., Qin, G., Mei, Y., & Sun, J. (2024). ImputeFormer: Low Rankness-Induced Transformers for Generalizable Spatiotemporal Imputation. KDD 2024.
>
> [4] Thangjam A, Jaipuria S, Dadabada P K. Time-varying approaches for long-term electric load forecasting under economic shocks[J]. Applied Energy, 2023, 333: 120602.

---

> ### Comment · Reviewer_xbnY · 2024-08-10
>
> Thanks for your detailed response and new experiments. I still have a few concerns.
>
> Regarding W1: I revisited the proof of Lemma 1. Unfortunately, Lemma 1 does not seem to be valid. The proof of Lemma 1 needs $\\epsilon\\to 0$. However, given two functions $f$ and $g$, the assumption that $\\|f(x)-g(x)\\|\\le\\epsilon$ implies $\\epsilon\\ge\\sup_x\\|f(x)-g(x)\\|$. As long as $f$ and $g$ are not identical, then $\\sup_x\\|f(x)-g(x)\\|>0$. This means that $\\epsilon$ cannot be arbitrarily small, falsifying the proof. Please either correct or remove Lemma 1, as wrong theoretical results should not be published. This is now my biggest concern.
>
> Besides the incorrect Lemma 1, the three points you mentioned seem to be trivial extensions of [28]. Although these ideas are somewhat new in time series imputation evaluation, their perspective novelty is rather limited. For instance, evaluating the performance via downstream tasks is the current mainstream for pretrained language or vision models. Furthermore, none of the three points is of technical novelty as your proposed method is basically [28]. This work seems to be just treating the "interpretation" results as "evaluation" results.
>
> Regarding L1: Thanks for your detailed explanation. I agree that there have been related empirical evidence, but my point is that it is difficult to evaluate the goodness of approximation. [28] showed that different datasets need different kernels, so it would be difficult to choose an appropriate kernel for a new dataset in practice when we do not have ground truth data. I am not blaming the proposed method; just pointing out a general limitation of RKHS approximation.
>
> Regarding L2: The examples you mentioned are irrelevent to the evaluation of time series imputation methods. Could you provide an example about how the fine-grained information will be used to evaluate time series imputation methods without using an aggregated metric?
>
> Regarding W2 & L3: Your responses sound reasonable to me. Thanks for your detailed elaboration. Please discuss about them in the paper.

---

> ### Author Response · Authors · 2024-08-10
> **Responses to the concerns of reviewer xbnY**
>
> We greatly appreciate the question raised, which is very important for us to improve the quality of our work.
> > Regarding W1
>
> 1. Proof of Lemma1
>
> We sincerely apologize for the confusion caused and acknowledge that in real-world applications, $\epsilon$ cannot be arbitrarily small. However, the proof of Lemma 1 does not necessitate $\epsilon$ to be arbitrarily small. Instead, it demonstrates that for any given $\delta$ and $\epsilon_2$, there always **exists** $\epsilon$ (not necessarily 0) that satisfies the condition.
>
> We suspect this misunderstanding might stem from our proof process. Let's revisit it. In the proof of Lemma 1, found in the appendix, we use $\epsilon \rightarrow 0$ to demonstrate the invalidity of the inequalities $|x-y| \delta \leq \epsilon$ and $N\left|\frac{2 \epsilon}{\delta}\right| \geq \epsilon_2$, thereby proving Lemma 1.
>
> However, the validity of these two inequalities also depends on $\epsilon_2$ and $\delta$. In other words, we only need to ensure that $\epsilon < |x-y| \delta$ and $|\epsilon| <\left|\frac{\epsilon_2 \delta}{2N}\right|$. It is important to note that $\delta$ and $\epsilon_2$ are set by us. If $\delta$ and $\epsilon_2$ are not extremely small, there is no need for $\epsilon$ to be extremely small.
>
>
> In practical applications, we don't need to require $\delta$ and $\epsilon_2$ to be sufficiently small since we only need to compare which of the imputation methods is more beneficial for downstream forecasting tasks at each time step. In this case, we only need to know the sign of the first derivative rather than an absolute accurate approximation. Both Table II in the original paper and our newly uploaded Table I in the attached PDF can prove the feasibility of doing so in practical operation.
>
> 2.
>
> We acknowledge that our approach shares some similarities with [28]. However, the method in [28] **can not be directly** used to solve our problem. We reiterate our differences, that is, their approach focuses on the importance of training data, while we evaluate the benefits of replacing one imputation value with another in the time series for downstream tasks.
>
> To our knowledge, we are the first to propose such a viewpoint and provide practical approaches as the work related to the pre-trained model is simply evaluated using some metrics of downstream tasks. On the contrary, we provide a rigorous mathematical definition of our task (see equation 1 in the original paper, which is a bi-level optimization problem) and conduct fine-grained evaluations.
>
> In addition, **our method can also be applied to combine different series imputation methods to bring benefits to downstream forecasting tasks.** These are properties that the relevant methods do not possess and can not be easily transferred. In addition, we also provided an **accelerated calculation** method in Section 2.3 in the original paper, which plays an important role when the model parameters and the amount of data increase(see table in Regarding L1).
>
> >Regarding L1
>
> We acknowledge that there are general limitations to using RKHS approximation. After our experiments on the original ELE dataset with 15-minute resolution (about 100000 training samples and the predicted length is 96), the NTK kernel achieved a good balance between computational efficiency and accuracy compared to methods like the influence function in our task. (with the **acceleration calculation** in Section 2.3).
>
> |   ELE   |  Mean | SAITS | Mean+SAITS+IF | Mean+SAITS+ours | Mean+SAITS+ours_seg16 | Mean+SAITS+ours_seg8 |
> |:-------:|:-----:|:-----:|:-------------:|:---------------:|:---------------------:|:--------------------:|
> |   MSE   | 0.249 | 0.307 |     0.248     |      0.238      |         0.236         |         0.237        |
> | Time(s) |   -   |   -   |     126.33    |     1053.28     |         264.27        |        118.53        |
>
> In addition, unlike [28], which focuses on explaining model outputs and necessitates high fitting accuracy, our method does not require such precision to deliver benefits, highlighting a key difference between the two approaches.
>
> >Regarding L2
>
> We are delighted to provide some relevant examples. The first example is that the importance is related to the value.  Suppose we have 10 generators, each with a power generation capacity of 10MW, totaling 100MW. Then filling 99.97MW as 100.02MW at the peak and filling 5MW as 5.05MW at the valley will result in the same aggregated error. However, due to the limitations of the generator, a 0.05MW error at the peak will require us to add a new generator to ensure power supply, resulting in significant additional economic expenses. The 0.05MW at the valley has no such concern. The second example considers the importance related to time or external events. When measuring a patient's blood pressure around the clock, the same margin of error in aggregated metrics can have different effects at different times or when patients engage in different physiological activities.

---

> > ### Comment · Reviewer_xbnY · 2024-08-10
> >
> > Thanks for your quick response!
> >
> > Regarding the proof of Lemma 1: Sorry for the confusion. Let me elaborate my concern more. To correct the proof, you need to prove that the $\\epsilon$ in the contradiction is indeed $\\ge\\sup_x\\|f(x)-g(x)\\|$ as this is your assumption.
> >
> > Regarding novelty: I agree that this work provides a new perspective in imputation evaluation. However, my point is that it is a trivial combination of two ideas (downstream task evaluation + interpretation). I believe that this would be an interesting work if it were submitted to the NeurIPS Datasets & Benchmarks track, but I expect more technical novelty in the main track.
> >
> > Regarding L1: Sorry for the confusion. What I wanted to emphasize is that "it would be difficult to choose an appropriate kernel for a new dataset in practice." You do not have to reply to this one. As this limitation is not a limitation of your method, my evaluation of your paper was not considering this limitation. Instead, you should discuss this limitation in the paper.
> >
> > Regarding L2: Sorry for the confusion. The examples you gave are not evaluating imputation methods; your examples are just evaluating each imputed value. In practice, merely evaluating each imputed value is pointless. When people imputation the time series, there is no way to evaluate because future data are not available. Thus, what we really need is to find out the best imputation method to use instead of just how good each imputed value is.

---

> ### Author Response · Authors · 2024-08-11
> **Responses to the concerns of reviewer xbnY(I)**
>
> Thank you again for the quick feedback.
>
> >Regarding the Lemma 1
>
> Sorry for the misunderstanding again. What we describe in Lemma1 is that ''$\||f(\boldsymbol{x})-g(\boldsymbol{x})\||$ is always less than $\epsilon$''. For continuous functions defined in bounded region, $\epsilon$ here is equal to $\sup_x\\|f(\boldsymbol{x})-g(\boldsymbol{x})\||$ and must exist (can not $\rightarrow$ $\infty$). This is **not the assumption** but the **definition**. **We did not make any assumption to let $\epsilon \geq \sup_x\\|f(\boldsymbol{x})-g(\boldsymbol{x})\||$ but we use $\epsilon$ to represent $\sup_x\\|f(\boldsymbol{x})-g(\boldsymbol{x})||$**. Therefore, we do not need to prove that $\epsilon \geq \sup _x\||f(\boldsymbol{x})-g(\boldsymbol{x})\||$ but rather examine the range of $\epsilon$ when Lemma1 holds for any given $\delta$ and $\epsilon_2$.
>
> Obviously, when $\epsilon$ can be arbitrarily small, the conclusion naturally holds. However, even though $\epsilon$ is not arbitrarily small, as we mentioned above, we only require that $\epsilon<|x-y| \delta$ and $|\epsilon|<\left|\frac{\epsilon_2 \delta}{2 N}\right|$. As long as the $\delta$ and $\epsilon_2$ we set are sufficiently large (even $\rightarrow$ $\infty$), these two inequalities are bound to hold, and the cost we pay is that our fitting accuracy will decrease. For this case, we have already demonstrated the feasibility of our method through a large number of experiments, because compared to [28], our task does not require particularly high fitting accuracy.
>
> >Regarding novelty
>
> We don't agree with the point that our work can be simply categorized as the combination of the downstream task evaluation + interpretation. In fact, we only used a similar RKHS approximation technique to [28] (RKHS approximation is not equal to interpretation since this technique was first proposed by [29]. However, this does not mean that [28] lacks technological innovation, as they used this technique to solve new problems and properly handle issues that would arise from trivial applications).
>
> In our case, it is **impossible** to obtain gradients by applying the RKHS approximation technique in a trivial manner, which has been carefully analyzed in our paper. This is also why we need to change the order of gradient calculation and approximation and analyze the rationality of doing so using Lemma 1. In addition, due to the need to consider the length of multi-step outputs in our method, the computational complexity will be high. For such cases, we also emphasize that we have provided the method in section 2.3 to **alleviate the computational burden** to make the approach practical in real applications.
>
> [28] C.-P. Tsai, C.-K. Yeh, and P. Ravikumar, “Sample-based explanations via generalized representers,” Advances in Neural Information Processing Systems, vol. 36, 2024.
>
> [29] B. Schölkopf, R. Herbrich, and A. J. Smola, “A generalized representer theorem,” in International conference on computational learning theory. Springer, 2001, pp. 416–426.

---

> ### Author Response · Authors · 2024-08-11
> **Responses to the concerns of reviewer xbnY(II)**
>
> >Regarding L2
>
> We do not agree that evaluating **each imputed value** is pointless. We can only consider an imputation method to be completely superior to the comparison method if it outperforms the comparison method at every time step. However, in reality, such a situation is almost impossible to exist. From this point of view, we will never find the real best imputation.
>
> 1. The importance of the fine-grained information
>
> Let's take the first example mentioned as an example and further extreme it. Suppose that imputation method **A** only fills 99.97MW as 100.01MW while imputation method **B** only fills 5MW as 5.05MW. The rest imputation values are the same, so from the perspective of aggregated metrics, **A** will be better than **B** because the error of 0.05MW is greater than 0.04MW. However, **A** will bring much greater economic losses than **B**. In this situation, we clearly can not conclude that method **A** is superior to method **B**.
>
> As for the example of measuring blood pressure in patients, it is even more convincing. If evaluating at every point is not important, why do doctors still need to design round-the-clock checks? **Obviously, their goal is not to calculate the patient's average blood pressure throughout the day.** Now since **knowing every time step in the measurement is important**, then there is no reason to claim that evaluating each imputed value is pointless.
>
> 2. The future data issues and why we need forecasting tasks as supervision
>
> When evaluating imputation, a portion of the future values can be obtained through observation over a while in the future. What truly cannot be obtained is the true label of the missing value, as it occurred in the unchangeable past. Therefore, the existing evaluations can only be used on simulated data, using aggregated metrics, which is truly deviating from real applications. Our approach of using real labels observed in the future (not simulated) and forecasting tasks as supervision is the evaluation method that can be used in real missing scenarios. Furthermore, our method can also combine existing advanced methods to achieve better effects than a single method.

---

> ### Comment · Reviewer_xbnY · 2024-08-11
> **Counterexample of Lemma 1**
>
> Regarding Lemma 1: You cannot let $\delta$ and $\epsilon_2$ be arbitrarily large because $\delta$ and $\epsilon_2$ are **given** (according to your statement of Lemma 1). If $\delta$ and $\epsilon_2$ are allowed to be large, then your Lemma 1 becomes trivial and insufficient to support the design of your method.
>
> If you still don't see the issue, please check your Lemma 1 with the following counterexample. Let the domain be $D=[0,1]$, let the two functions be $f(x)=0$ and $g(x)=\epsilon\sin\big(\frac{\pi x}{\min\\{\epsilon,1\\}}\big)$, and consider $\delta=0.5$, $\epsilon_2=0.3$. It is easy to see that $f$ and $g$ are infinitely differentiable w.r.t. $x$, and that $\sup_x|f(x)-g(x)|=\epsilon$. However, no matter what $\epsilon$ is, the measure of the region where $|f'(x)-g'(x)|>\delta$ is at least $\frac23\min\\{\epsilon,1\\}\big\lfloor\frac1{\min\\{\epsilon,1\\}}\big\rfloor\ge\frac13>\epsilon_2$. (A similar counterexample can be constructed for other given $\delta$ and $\epsilon_2$ as well.)

---

> > ### Author Response · Authors · 2024-08-11
> > **Responses to the concerns of reviewer xbnY**
> >
> > Thank you again for your quick feedback.
> >
> > >Regarding the counterexample
> >
> > Thank you very much for providing the counterexample. However, we are sorry to say that the example you provided does not seem to meet most scenarios in real application and we will explain why in the following.
> >
> >  In section 2.2 of the original paper, we state that our goal is to calculate $\frac{\partial f\left(X_k^v, \theta\right)}{\partial y_{i, l}}$, which is a symbolic expression and can be conceptually broken down into $\frac{\partial f\left(X_k^v, \theta\right)}{\partial \theta} \cdot \frac{\partial \theta}{\partial y_{i, l}}$, elucidating the role of the label $y_{i, l}$ in shaping the model parameters $\theta$ throughout the training process. This, in turn, has repercussions on the model's prediction when evaluated on unseen data from the test set, i.e. $f\left(X_k^v, \theta\right)$. Therefore, our goal is actually to estimate the impact of label $y_{i,l}$'s changes on the model parameter $\theta$. As we mentioned in the **L1 of rebuttal**, for widely used loss functions $\mathcal{L}$ like MSE,  $\frac{\partial^2 \frac{\partial \mathcal{L}\left(f_\theta(\boldsymbol{X_i}), \boldsymbol{y_i}\right)}{\partial \theta}}{\partial y_{i, l}^2}=0$, which indicates that the second derivative of the $y_{i,l}$ should make no effect on the final result. **In the example the reviewer provided, the second derivative of $x$ is not equal to zero, which does not meet most scenarios.**
> >
> > During the above process, Lemma1 plays the role that allows us to change our goal from directly approximate $\frac{\partial f\left(X_k^v, \theta\right)}{\partial y_{i, l}}$ to approximate $f\left(X_k^v, \theta\right)$ first and then calculate derivatives rather than quantitative analysis. However, in Lemma1, we did not provide corresponding restrictions to obtain a generalized situation, which comes at the cost that in some cases, Lemma1 may require a relatively large $\epsilon_2$ and $\delta$, making our restrictions relatively relaxed. We apologize again for the misunderstanding caused.

---

> ### Comment · Reviewer_xbnY · 2024-08-11
>
> I am just saying that your Lemma 1 is mathematically wrong (NOT saying that it is wrong in real-world scenarios because real-world scenarios have way more underlying assumptions than your Lemma 1). Basically, if you require that the second derivative is zero and that the function is infinitely differentiable, then the function is necessarily a quadratic (or linear or constant) function. In this case, Lemma 1 becomes trivial.
>
> As your proof of Lemma 1 does not use any assumption other than those in the statement of your Lemma 1, my counterexample still applies to your Lemma 1. If you still don't see the issue, please give your "relatively large" $\epsilon_2$ and $\delta$ where you believe your Lemma 1 holds, and I will construct the corresponding counterexample.

---

> ### Author Response · Authors · 2024-08-12
> **Responses to the concerns of reviewer xbnY**
>
> >Regarding the correctness of Lemma 1
>
> We apologize for any misunderstandings caused. Perhaps we should re-examine Lemma 1 from two perspectives: mathematical proof and practical application.
>
> **From a mathematical proof perspective:**  We confirm that Lemma1 is mathematically correct because we are proving the existence of $\epsilon$ in Lemma1. In fact, in the example you provided, $\epsilon = 0$ can satisfy all the values of $\epsilon_2$ and $\delta$. The existence of $\epsilon$ that meets the conditions has not been challenged. Therefore, Lemma1 does not have any mathematical errors.
>
> **From a practical application perspective:** In fact, we have not claimed that we can use Lemma1 for specific quantitative analysis, and we also acknowledge that $\epsilon$ is difficult to equal to 0 in practical applications. Lemma 1's role here is to provide us with the rationality to simplify the problem (note that rationality does not necessarily mean 100% accuracy, analogical linear approximation can introduce significant errors in some cases, but it is still widely used for simplifying problems). The proof process of Lemma1 tells us that $\epsilon$ does not necessarily need to be constant at 0. Therefore, in practical situations, we can consider using this to simplify the problem.
>
> **We apologize again for making you feel that we are ignoring the issue and we sincerely appreciate the efforts you paid to help us.** However, the issue actually did not challenge the mathematical correctness of Lemma1. If so, you should provide a counterexample that Lemma1 does not hold when $\epsilon = 0$ (at which point $\epsilon$ that meets the condition does not exist). On the contrary, **your issue is challenging us for whether Lemma1 can help us in the real application(for example, in the counterexample you raised)**. Therefore, we have tried to explain to you our rationality in practical application.
>
> >Detailed analysis in the application and modification of Lemma1.
>
> Here we would like to explain to you in more detail why your example is not seen in most of the cases. In fact, the conclusion cannot be obtained trivially, because it is related to the loss function and optimizer we choose during training. We need to analyze the specific situation. Here we take the widely used MSE loss function and two optimizers SGD and ADAM as examples and we set learning rate as $\eta$.
>
> Remind that our goal is to calculate $\frac{\partial f\left(X_k^v, \theta(y_{i,l})\right)}{\partial y_{i, l}}$, where $\theta(y_{i,l})$ is the model parameters and related to $y_{i,l}$. Let $g_t(y_{i,l}) = \frac{ \partial \mathcal{L_t} (f_\theta(X),y)}{\partial \theta}$. Note that $\frac{\partial^2 g_t(y_{i,l})}{\partial y_{i,l}^2} = 0$ for MSE loss function. For SGD optimizer, $\theta_{t} = \theta_{t-1}-\eta g_t(y_{i,l})$. Therefore, $\theta_T(y_{i,l}) = \theta_0 -\sum_{t=1}^{T} \eta g_t(y_{i,l})$, where T is the training epoches. In this case, $\frac{\partial^2 \theta_T(y_{i,l})}{\partial y_{i,l}^2} = -\sum_{t=1}^{T} \eta\frac{\partial^2 g_t(y_{i,l})}{\partial y_{i,l}^2} = 0$ and we only need to consider the linear case. While with the ADAM optimizer, things will be a little bit complicated. For simplity, let $\theta_T(y_{i,l}) = \theta_0 -\sum_{t=1}^T\eta v(g_t(y_{i,l}))$, where v() represents the terms in ADAM. Then $\frac{\partial^{k+2} \theta(y_{i,l})}{\partial y_{i,l}^{k+2}} = -\sum_{t=1}^T\eta \frac{\partial^k(\frac{\partial^2 v}{\partial g^2}(\frac{\partial g}{\partial y_{i,l}})^2+\frac{\partial v}{\partial g}\frac{\partial^2 g}{\partial y_{i,l}^2})}{\partial y_{i,l}^k}$($k \geq 1$), which is $O(y_{i,l}^{\frac{k+1}{2k+1}})$ and tends to 0 as $k \rightarrow \infty$. The counterexample you provide is not this kind of case, that is, the derivatives of g(x) is not in the form of $O(x^{\frac{k+1}{2k+1}})$.
>
> In the subsequent changes, we will consider incorporating these common situations into Lemma1's constraints and analyzing the changes in errors under different conditions. Thank you again for your help.

---

> > ### Comment · Reviewer_xbnY · 2024-08-12
> >
> > $\epsilon=0$ means that $f=g$ everywhere. If you allow $\epsilon=0$, then Lemma 1 holds trivially with $f=g$, and there would be nothing that needs you to prove.
> >
> > I agree that your intuition helps you design your method, but intuition is not a mathematical proof. If you just want to emphasize your intuition, then do not state it as a lemma. Otherwise, if you decide to state it as a lemma, I will have to ensure it to be a rigorous mathematical statement.

---

> ### Author Response · Authors · 2024-08-12
> **Responses to the reviewer xbnY**
>
> Dear reviewer xbnY
>
> Based on your suggestion, we will make some modifications to the relevant wording. Thank you again for taking the time to provide valuable suggestions.

---

> ### Comment · Reviewer_xbnY · 2024-08-12
>
> Thanks for your understanding. Could you let me know how you plan to modify the wording? I need to make sure that it is a rigorous mathematical statement, or otherwise I cannot raise my score.

---

> > ### Author Response · Authors · 2024-08-13
> > **Responses to the reviewer xbnY**
> >
> > Dear reviewer xbnY
> >
> > Thank you very much for your reply. We are more than happy to provide you with our ideas for modifying the text and hope that you can provide some valuable suggestions, which will greatly help us continue to improve the quality of our work.
> >
> > Overall, based on the reasons you mentioned, we will define Lemma1 as a remark, which is to demonstrate the intuition of our work. The specific description will be highlighted below.
> >
> > >Remark 1. Given two infinitely differentiable functions $f(\boldsymbol{x})$ and $g(\boldsymbol{x})$ in a bounded domain $D \in R^n$. Let $\epsilon = sup_\boldsymbol{x}||f(\boldsymbol{x}) - g(\boldsymbol{x})||$. For any given $\delta$ and $\epsilon_2$, there exists  $\epsilon$ (not necessarily have to be 0 in some cases) such that, in the domain $D$, the measure of the region $I$ that satisfying $||\frac{\partial f(\boldsymbol{x})}{\partial \boldsymbol{x}} - \frac{\partial g(\boldsymbol{x})}{\partial \boldsymbol{x}}|| > \delta$ is not greater than $\epsilon_2$, i.e, $m(I)\leq \epsilon_2$.
> >
> > We will keep the derivation process (not proof) in the appendix and add some comments below.
> >
> > >As shown in the derivation process above, for some given $\delta$ and $\epsilon_2$, we only require that $|\epsilon|<|x-y| \delta$ and $|\epsilon|<\left|\frac{\epsilon_2 \delta}{2 N}\right|$. Even though the real boundary is highly related to the specific scenarios and is difficult to tell in real applications, we can have a look at some widely used examples.
> >
> > Then we will give some analysis of widely used cases for MSE loss functions with SGD and ADAM optimizers. Before, we want to analyze here why the counterexample seems to look like a contradiction to our remark.
> >
> > The fundamental reason for this is that in this example, (we consider f as 0), we need to consider a function h (x) whose derivative is a periodic function and **the number of monotonic intervals is infinite**, By adjusting the parameters, we can make the period length infinitely small. In this case, even if the derivative is large (a significant difference from the derivative of f), the function can still achieve a situation where it first increases and then decreases, ensuring that the values within the domain are small (i.e., the difference from f is small).
> >
> > > For the loss function $\mathcal{L}$ to be set as MSE. Note that we use $X$ and $X_{test}$ to represent the input data in the training set and test set, separately. $y$ is the training label and $y_{i,l}$ is the value of a time step in it. We use t to represent the training epoch and there will be T epochs total. Our goal is to use $g(X_{test},y_{i,l})$ to approximate $f(X_{test},\theta_T,y_{i,l})$ first and then use $\frac{\partial g(X_{test},y_{i,l})}{\partial y_{i,l}}$ to approximate $\frac{\partial f(X_{test},\theta_T,y_{i,l})}{\partial y_{i,l}}$.
> >
> > >During the training process of a forecasting model, we let $h_t(y_{i,l})$ represent $\frac{\partial \mathcal{L}(f(X,\theta_t,y_{i,l}), y)}{\partial \theta_t}$ and $\frac{\partial^2 h_t(y_{i,l})}{\partial y_{i,l}^2}$ will be zero. Recalling that our goal is to approximate the gradient $\frac{\partial f(X_{test},\theta_T,y_{i,l})}{\partial y_{i,l}}$ by approximating  $f(X_{test},\theta_T,y_{i,l})$ first and then take the gradient. However, the function that we really want to approximate is $\frac{\partial \theta_T}{\partial y_{i,l}}$ since $\frac{\partial f(X_{test},\theta_T,y_{i,l})}{\partial y_{i,l}} = \frac{\partial f(X_{test},\theta_T,y_{i,l})}{\partial \theta_T}\frac{\partial \theta_T}{\partial y_{i,l}}$ and $\frac{\partial f(X_{test},\theta_T,y_{i,l})}{\partial\theta_T}$ is a constant for given $X_{test}$.
> >
> > >For SGD (as well as its variants SGD with momentum), $\theta_T=\theta_0-\sum_{t=1}^T \eta h_t\left(y_{i, l}\right)$. Therefore, the $\frac{\partial^2 \theta_T}{\partial y_{i,l}^2} = 0$, which means that the gradient of the function we want to approximate is constant. In this case, our approximation will be pretty good.
> >
> > >For ADAM, on the one hand, [1] has claimed that the first-order approximation in the SGD situation remains valid, as long as a small step-size $\eta$ is used in the update. On the other hand, let $\theta_T=\theta_0-\sum_{t=1}^T \eta v_t(h_t\left(y_{i, l}\right))$, where $v_t$ represent the terms in ADAM. In this situation, $\frac{\partial \theta_T}{\partial y_{i,l}}$ will be an algebraic function with only **a finite number of monotonic intervals**. Therefore, for any given $\delta$ and $\epsilon$, the $\epsilon_2$ will not be really high since the $y_{i,l}$ that makes $||\frac{\partial f(X_{test}, \theta_T, y_{i,l})}{\partial y_{i,l}} - \frac{\partial g(X_{test}, y_{i,l})}{\partial y_{i,l}}|| > \delta$ can only appear near the local extreme point, whose number is finite.
> >
> > [1] Pruthi G, Liu F, Kale S, et al. Estimating training data influence by tracing gradient descent[J]. Advances in Neural Information Processing Systems, 2020, 33: 19920-19930.

---

> > ### Author Response · Authors · 2024-08-13
> > **Responses to the reviewer xbnY**
> >
> > Dear reviewer xbnY
> >
> > Please note that the above is only a draft that we have revised. If you have any comments, please let us know.

---

> ### Comment · Reviewer_xbnY · 2024-08-13
>
> Thanks for your update. My concern about Lemma 1 has been addressed. One minor issue is that the current Appendix B still "looks like" a proof. To avoid misleading the reader, please remove "contradiction" (because there is no contradiction in the counterexample) and "countable" (because you only discussed the case where $N$ is finite) and focus on the construction of the intervals instead.
>
> I have raised my score from 3 to 4. To help me decide whether to further increase my score, let's continue to discuss about my remaining concerns.
>
> Regarding the future data issue: In your examples for L2, you mentioned that one can observe some future values and use them for evaluation. My concern is that near-future observations might not provide much information for evaluation because most forecasting methods should be very good at predicting near-future values. Consequently, one may need to acquire distant-future values (i.e., a large number of future values) to obtain sufficient information for evaluation.
>
> Regarding Sec 2.3: You claimed that $A^\dagger Af\approx f$. However, this is not necessarily a good approximation because the pseudo-inverse in general only ensures that $A(A^\dagger A)=A$, that $(A^\dagger A)A^\dagger=A^\dagger$, and that $A^\dagger A$ is Hermitian but does not ensure that $A^\dagger A$ is close to the identity matrix. The difference between $A^\dagger A$ and the identity matrix depends on the specific matrix $A$. For the matrix $A$ given in the paper, do you have any theoretical guarantee about this approximation?

---

> ### Author Response · Authors · 2024-08-13
> **Responses to the reviewer xbnY**
>
> Dear reviewer xbnY
>
> Thank you very much for your quick feedback and we will address your concerns below.
>
> >Regarding the "proof"
>
> Thank you for your suggestion. We will continue to adjust the specific text, with the main purpose of explaining the motivation and rationality of our method rather than proof.
>
> >Regarding future data issues.
>
> We acknowledge that the quantity of future data does have an impact on the evaluation. Theoretically, more future data is more likely to provide better evaluation criteria.
>
> However, this does not mean that our method is difficult to implement when the amount of future data is small. We add a simple experiment based on the ELE dataset. We first adjusted the task to predict the value of the next 4 hours. Then we set the training set to **3 months**, the validation set for evaluation to **1 day**, and then set the test set to the next **3 months**. The rest are the same as the setting in the original paper. The specific experimental results are shown in the table below.
> |  MSE  |  Mean |       SAITS      |     BRITS    |     GPVAE    |     USGAN    | ImputeFormer |
> |:-----:|:-----:|:----------------:|:------------:|:------------:|:------------:|:------------:|
> |   -   | 0.853 |   0.804(0.005)   | 0.854(0.001) | 0.865(0.006) | 0.894(0.017) | 0.839(0.017) |
> | +Ours |   -   | **0.784(0.000)** | 0.796(0.000) | 0.797(0.004) | 0.807(0.003) | 0.793(0.005) |
>
> Even in such an extreme environment (the validation set that can be used for evaluation is only **1 day**, and the test set is used to show whether the evaluation is reasonable), our method can still help the forecasting model achieve better results, that is, obtain imputation values ​​that are more conducive to downstream forecasting tasks.
>
>
>
>
>
> >Regarding the section 2.3
>
> Yes, we concur that $\boldsymbol{A}^{\dagger}\boldsymbol{A}$ is not close to the identity matrix. Indeed, $\boldsymbol{A}^{\dagger}\boldsymbol{A}$ acts as a projection matrix, where $\boldsymbol{A}^{\dagger}\boldsymbol{A}f(x)$ projects the values of $f(x)$ onto the row space of $\boldsymbol{A}$. The fitting error of $\boldsymbol{A}^{\dagger}\boldsymbol{A}f(x)$ relative to $f(x)$ in the $\ell_2$ norm equals the value of $\min_{\boldsymbol{\alpha}}\|f(x)-\boldsymbol{A}^T\boldsymbol{\alpha}\|_2$. To see why this is so, without loss of generality, let us assume that $\boldsymbol{A}$ is a full row-rank matrix. The $\boldsymbol{\alpha}$ that minimizes $\|f(x)-\boldsymbol{A}^T\boldsymbol{\alpha}\|_2$, denoted as $\boldsymbol{\alpha}^{\star}$, is given by $(\boldsymbol{A}\boldsymbol{A}^T)^{-1}\boldsymbol{A}f$. This implies that $\boldsymbol{A}^T\boldsymbol{\alpha}=\boldsymbol{A}^T(\boldsymbol{A}\boldsymbol{A}^T)^{-1}\boldsymbol{A}f=\boldsymbol{A}^{\dagger}\boldsymbol{A}f$. Therefore, we have $\|f(x)-\boldsymbol{A}^T\boldsymbol{\alpha}^{\star}\|_2=\|f(x)-\boldsymbol{A}^{\dagger}\boldsymbol{A}f\|_2$. Based on this insight, we understand that the fitting error of $\boldsymbol{A}^{\dagger}\boldsymbol{A}f(x)$ depends on the design of $\boldsymbol{A}$.
>
> Given that $f(x)$ represents a time series, we can leverage properties of time series to design $\boldsymbol{A}$, such as sparsity in the frequency domain or smoothness. Here, we primarily consider $f(x)$ as a smooth function that can be approximated by a piece-wise constant function. Thus, designing $\boldsymbol{A}$ in a form as presented in the paper, the fitting error equates to using an optimal piece-wise constant function to approximate $f(x)$. Since $f(x)$ is typically quite smooth, the fitting error tends to be relatively small.

---

> > ### Comment · Reviewer_xbnY · 2024-08-13
> >
> > Thanks for your detailed elaboration! I have some follow-up questions.
> >
> > Regarding the future data issue: Thanks for your new experiment. To my understanding, the ELE dataset has a relatively high frequency (15 min). Thus, a day gives 96 new observations, so it should be fine to use a day as the validation set. However, it might be time-consuming to obtain so many new observations for low-frequency time series. Hence, your method seems to be more suitable for high-frequency data than for low-frequency data. Could you discuss this limitation in the paper?
> >
> > Regarding Sec 2.3: Thanks for your explanation. If there is no theoretical guarantee, could you add an ablation study to empirically compare the error of this approximation between non-smooth and smooth time series? For non-smooth time series, you may use the fractional Brownian motion with Hurst parameter > 1/2 (say, 0.75).

---

> > > ### Author Response · Authors · 2024-08-13
> > > **Responses to the reviewer xbnY**
> > >
> > > Dear reviewer xbnY
> > >
> > > Thank you for your feedback, and we would like to explain the two issues below.
> > >
> > > >Regarding the future data issue
> > >
> > > Sorry for the misunderstanding, the ELE dataset here we used is in hourly resolution [1](also cited as [34] in the original paper). Therefore, there are just 24 data points in a day. However, as you mentioned, compared to low-frequency datasets, our method has better application scenarios in high-frequency datasets because it is more conducive to obtaining more data. Therefore, we will add relevant discussions in the revised version.
> > >
> > > >Regarding the Sec 2.3
> > >
> > > We use the following Python code to generate the series you mentioned.
> > > ```python
> > > from fbm import FBM
> > > f = FBM(n=2281, hurst=0.75, length=1, method='daviesharte')
> > > ```
> > > Then we use the GEF and ELE datasets to act as the smooth time series(we did not add missing value here). Note that we set the number of training data to 2186(about 3 months), the validation data to 96(3 days), and the predicted length to 24. Then we consider the calculation of the original approach as the true label and calculate the loss between it and the calculation of our approach using the acceleration method in Sec 2.3. The following table shows the error between the accelerated results and the original results under different numbers of segments. The brackets indicate how much the error under this segment increases compared to the error of Seg12.
> > >
> > > |  MSE  |         GEF        |         ELE        |       Brownian      |
> > > |:-----:|:------------------:|:------------------:|:-------------------:|
> > > | Seg12 |      7.0735e-9     |      1.3408e-9     |      3.74636e-6     |
> > > |  Seg8 | 7.0737e-9($\uparrow$0.0028%) | 1.3410e-9($\uparrow$0.0014%) | 3.74639e-6($\uparrow$0.0008%) |
> > > |  Seg6 | 7.0739e-9($\uparrow$0.0056%) | 1.3411e-9($\uparrow$0.0022%) | 3.74640e-6($\uparrow$0.0010%) |
> > > |  Seg4 | 7.0741e-9($\uparrow$0.0084%) | 1.3412e-9($\uparrow$0.0029%) | 3.74644e-6($\uparrow$0.0021%) |
> > >
> > > It can be seen that as the number of segments increases, the error also increases, but the magnitude is very small, including the generated Brownian motion data. This shows that our acceleration method can speed up the calculation of our method with almost no loss of accuracy.
> > >
> > >
> > >
> > > [1] G. Lai, W.-C. Chang, Y. Yang, and H. Liu, “Modeling long-and short-term temporal patterns with deep neural networks,” in The 41st international ACM SIGIR conference on research & development in information retrieval, 2018, pp. 95–104.

---

> > > > ### Comment · Reviewer_xbnY · 2024-08-13
> > > >
> > > > Thanks for your quick response.
> > > >
> > > > Regarding your new experiment, it seems that FBM does have higher error than GEF and ELE, validating your hypothesis that the proposed matrix $A$ is more suitable for smooth $f$. Please discuss this limitation in the paper. By the way, are the results absolute or relative errors? If they are absolute errors, please also include relative error results into your paper as true gradients are typically small as well.
> > > >
> > > > My concerns have been addressed, so I have raised my score. Please revise the paper according to our discussion above.

---

> > > > > ### Author Response · Authors · 2024-08-14
> > > > > **Responses to the reviewer xbnY**
> > > > >
> > > > > Dear reviewer XbnY
> > > > >
> > > > > Thank you very much for your recognition of our work. We will organize the relevant discussions during this period and incorporate them into our revised version. Thank you again for taking the time to provide us with valuable suggestions.

---

### Official Review · Reviewer_56K9 · 2024-07-07

**Soundness:** 2
**Presentation:** 3
**Contribution:** 3
**Rating:** 7
**Confidence:** 4

**Summary:**

The authors propose a strategy that evaluates the effectiveness of various imputation methods used to fill missing values ​​at different timestamps (time series). The effectiveness of each imputation method is evaluated based on the downstream task gain. Subsequently, rather than filling missing values with a single imputation method, each missing value at a specific timestamp is filled with the most accurate imputation method at that timestamp.

**Strengths:**

- Originality: The originality of this work lies in the fact that, unlike previous works that study the impact of missing value in inputs, this study evaluates the impact of missing value in labels (time series). Moreover, it proposes an alternative solution that does not use a single imputation method but several that maximize the gain of the downstream task by providing the most accurate imputed value at each timestamp;

- Quality: The paper quality is good. The document is well-structured;

- Significance: The reviewer believes that the proposed solution could have a significant impact in the community. Indeed, one of the inherent problems with time series data is missing values. The proposed solution allows for an optimal choice of various imputation models or strategies to fill these missing values. Moreover, their solution is time series forecasting task-agnostic.

**Weaknesses:**

- The experimental section 3.3.1 is unclear and very difficult to follow. Authors must refer to the questions to know the reviewer's concerns;

- The proposed strategy may fail when the labels are multivariate time series (MTS). Indeed, in the case of MTS, many imputation methods/models (e.g. deep learning models) consist of filling missing values based on all observed values, i.e. within and across streams. Consequently, combining these imputation methods/models with others that fill in missing values by considering only the observed values of the corresponding univariate time series (e.g. the empirical mean) can lead to a high variance in the imputed values. This high variance will introduce noise into the model calculation scheme and may affect the gain of the downstream task;

- Using the mean as the sole baseline is not sufficient. Authors should adopt at least two baseline imputation methods to perform robust experiments;


- The limitations of the model are not discussed. The authors simply indicate the directions for future work.

**Questions:**

- Can you please formally detail how you go from the indicator function $I(i,l)$ (equation 1) to its approximation with the first order Taylor expansion (equation 2)? It is quite confusing because in the equation we have the losses with respect to the learnable parameters $\theta_1$ and $\theta_2$ and in Equation 2 only the loss with respect to the $\theta$ parameter is used. Also, where is the initial term of the first order Taylor expansion? Did you just assume that it is equal to zero?

- Shouldn't NTK be equal to $\sum_{p=1}^{P}\frac{\partial f (\mathbf{X}_i, \theta)}{\partial \theta_p}\frac{\partial f (\mathbf{X}_k^v, \theta)}{\partial \theta_p}$ in Equation 8?

- Can you please clarify for what purpose linear interpolation is used (line 259-261)? Is it to fill in missing values ​​in the labels and thus obtain two labels, one filled in by the empirical mean and the other filled in by this interpolation?

- Have you really retrain 8760 forecasting models (line 263), or is this a typing error?

- Does the actual gain represent the gain obtained using a single imputation method, and does the estimated gain represent the gain that could be obtained using a combination of imputation methods? If not, can you explain these gains?

- Can you please elaborate more about the influence function?

- When evaluating the proposed solution with multivariate time series (MTS), are the labels also MTS, or are they still univariate time series?

**Limitations:**

- Unless the reviewer has missed something, it appears that the proposed method is only applicable when the labels are univariate time series. This makes it less generalizable, since in many real-world forecasting applications the objective is to forecast multivariate time series at consecutive discrete timestamps;

- The reviewer would suggest adding the computation time to obtain the estimation gain for each model in Table 2;

- Please correct the typo on line 139, Taylor expansion;

- The reviewer suggests that as a future work, the authors empirically evaluate the gain obtained by combining imputation methods/models that adopt the same or different approaches to filling missing values. For example, it would be interesting to see how the emprirical mean imputation works with GRU-D[1], which also uses the empirical mean to fill missing values. Another example could be to study the gain obtained when imputation models based on graph neural networks and message passing are used exclusively or combined with other imputation models, such as those based on recurrent neural networks.


[1] Z. Che, S. Purushotham, K. Cho, D. Sontag, Y. Liu, Recurrent neural networks for multivariate time series with missing values, Scientific reports 8 (1) (2018) 6085.

---

> ### Author Rebuttal · Authors · 2024-08-06
>
> We thank the reviewer for the valuable feedback and we have addressed each of the concerns raised by the reviewer as outlined below.
>
> >Weakness 1
>
> We sincerely apologize for the misunderstanding we caused, and we will make detailed modifications in the revised version.
>
> >Weakness 2 & Question 7 & Limitation 1
>
> The experiment in Table 3 uses a multivariate time series to forecast a univariate one. However, our method can theoretically be applied in multivariate forecasting, as in multi-step forecasting, the label $y$ itself is a multidimensional variable. We only need to make simple modifications to the code to apply our method. Below, we present a simple experiment to demonstrate the effectiveness of our method in multivariate forecasting. For simplicity (the main limitation here comes from the time consumption of the imputation method itself rather than our method), we predicted the first three sequences in the original ELE dataset.
>
> | Multi forecasting on ELE |   MSE  |
> |:-----------------:|:------:|
> |        Mean       | 0.1776 |
> |       SAITS       | 0.2252 |
> |     SAITS+ours    | **0.1497** |
>
> >Weakness 3
>
> Using a simple average as the baseline is convenient for evaluation, as each imputation method has different results under different random numbers, and using them as the baseline for evaluation will become very complex. However, to demonstrate the effectiveness of our method under different baselines, we validated it on ELE data using SAITS and ImputeFormer(IF). If the reviewer consistently believes in the necessity of multiple baselines, we will include them in the revised version.
>
> |    MSE       |     ELE    |
> |:------------------:|:----------:|
> |        Mean        |   0.1123   |
> |        SAITS       |   0.1176   |
> |         IF         |   0.1076   |
> | SAITS(baseline)+IT | **0.1013** |
> | IT(baseline)+SAITS |   0.1051   |
>
> >Weakness 4
>
> We discussed the limitations in the Appendix and we will add them to the conclusion in the revised version.
>
> >Question 1
>
> We apologize for the misunderstanding. Our goal is to obtain the value of $\theta_2$ without retraining given $\theta_1$ so that we can now the performance of the model on the parameter $\theta_2$ and get the gain for changing from $\theta_1$ to $\theta_2$. The $\theta$ in Eq (2) should be theta $\theta_1$ and we take $y_ {i, l} ^ {(1)} $ as the baseline and examine the benefits of replacing $y_ {i, l} ^ {(1)} $with $y_ {i, l} ^ {(2)} $. At this point, the init term is $y_ {i, l} ^ {(1)}$.
>
> >Question 2
>
> Due to the need for multi-step output in our network, the size of the two terms in the NTK kernel is ($L_2$, p) and (p, $L_2$), where $L_2$ is the length of the output. Therefore, the NTK kernel here is not a single value, but rather a matrix of size ($L_2$,$L_2$).
>
> >Question 3
>
> There is no particular reason for using linear interpolation here, mainly for the sake of simplicity. Our main goal is to demonstrate that our method can estimate the impact of replacing average interpolation with linear interpolation at each time step on downstream forecasting tasks. Replacing linear interpolation with other imputation methods will not bring about essential differences.
>
> >Question 4
>
> There is no error and we train 8760 different models so that we can get the real difference between the original model and the model that replaces the baseline value with the other ones during the training process, to test the effectiveness of our estimation. If necessary, we can put them on anonymous Google Drive after the rebuttal.
>
> >Question 5
>
> Suppose we have two imputation methods, one serves as the baseline and we want to combine the other one. The gain obtained by replacing the baseline value with the value from the other method at each time step is the actual gain, and the estimated gain is our method's estimation of the actual gain. Therefore, higher accuracy and correlation represent better performance.
>
> >Question 6
>
> The Influence function examines how the model changes on the test set when a training sample is missing during the training process, and uses this to explain the importance of the sample. Our method mainly examines how the model changes on the test set after a sample is transformed from one imputation method to another during the training process. The application methods of the two are also different. The former usually discards samples with negative impacts to seek better performance on the test set, while our method can evaluate which imputation method is used at each time point in the sample to perform better on the test set.
>
> >Question 7 & Limitation 1
>
> See Weakness 2
>
> >Limitation 2
>
> Here is the time consumption in Table 2 with RTX 3080Ti and the time consumed by our methods is very small after our optimized programming.
>
> | Time(s) |   GEF  |  ETTH1 | ETTH2  | ELE    | Traffic | AIR    |
> |:----:|:------:|:------:|--------|--------|---------|--------|
> |      | 19.908 | 20.233 | 20.908 | 30.170 |  29.132 | 14.701 |
>
> >Limitation 3
>
> Thank you for your pointing out and we will correct it and carefully correct the similar mistake in the revised version.
>
> >Limitation 4
>
> Thank you very much for your suggestion. We strongly agree that a single imputation method is difficult to apply in various scenarios. How to combine the advantages of different methods will be an interesting direction.

---

> > ### Comment · Reviewer_56K9 · 2024-08-12
> >
> > Dear authors, thank you for your detailed answers. My concerns were clarified. I keep the score unchanged.

---

> > > ### Author Response · Authors · 2024-08-12
> > > **Response to reviewer 56K9**
> > >
> > > Dear reviewer 56K9
> > >
> > > Thank you very much for your feedback. We are also pleased that our response has addressed your concerns. On behalf of all authors, I sincerely thank you for your recognition of our work and valuable suggestions.

---

### Official Review · Reviewer_z2pu · 2024-07-12

**Soundness:** 2
**Presentation:** 3
**Contribution:** 2
**Rating:** 4
**Confidence:** 4

**Summary:**

The paper proposes a task-oriented time series imputation evaluation approach that assesses the impact of different imputation strategies on downstream forecasting tasks, rather than just the accuracy of the imputed values. The authors introduce a similarity-based method to efficiently estimate the impact of imputed values on downstream task performance, and develop a time series imputation framework that combines the advantages of different imputation strategies. The paper also discusses several axioms that the proposed method satisfies, such as the efficiency, self-explanation, and symmetric zero axioms.

**Strengths:**

1. The paper proposes a task-oriented time series imputation evaluation method, focusing on evaluating the impact of different imputation strategies on downstream prediction tasks, rather than just evaluating the accuracy of imputed values. This approach is aligned with practical application needs.

2. The authors propose a similarity-based method that achieves a balance between performance and computational cost, and can effectively estimate the impact of imputed values on the performance of forecasting tasks.

3. The authors have developed a time series imputation framework that combines the advantages of different imputation strategies, and achieves better performance on downstream forecasting tasks.

**Weaknesses:**

1. The authors state in the abstract that missing values can be observed in many time series analysis tasks, but the task-oriented imputation evaluation method proposed by the authors is only designed for forecasting tasks. In the problem definition, the label of a time series dataset is similarly defined as a time series, however in the classification task, the label is a discrete value. In addition, for some unsupervised learning time series tasks, such as clustering [1], missing values can affect the performance as well. In conclusion, the authors' method is too limited in its scope of application compared to other imputation methods.

[1] Time series cluster kernel for learning similarities between multivariate time series with missing data. PR 2018.

2. The authors' motivation is to provide imputation values that are more beneficial to downstream forecasting tasks, while ignoring the accuracy of the imputation values themselves. However, for the time series imputation task, performing accurate imputations is also one of the non-negligible purposes [2]. The authors ignored the requirement for imputation accuracy when designing their methodology, and did not show experimentally whether their method would reduce imputation accuracy.

[2] Deep learning for multivariate time series imputation: A survey. Arxiv 2402.04059.

3. The authors' ultimate aim is to make more accurate predictions in the presence of missing values, which is highly relevant to the study of robust time series forecasting [3,4]. However the authors don't mention them in the paper and don't compare them as baselines in their experiments. In addition the authors should discuss the advantages and disadvantages of imputation-based and missing value robust-based forecasting methods.

[3] Weakly Guided Adaptation for Robust Time Series Forecasting. VLDB24.

[4] RobustTSF: Towards Theory and Design of Robust Time Series Forecasting with Anomalies. ICLR 24.

**Questions:**

Please see weaknesses.

**Limitations:**

Yes

---

> ### Author Rebuttal · Authors · 2024-08-06
>
> We thank the reviewer for the valuable feedback and we have addressed each of the concerns raised by the reviewer as outlined below.
>
> >Weakness 1
>
> We apologize for the misunderstanding caused by our statements. Our main focus here is on time series forecasting tasks. Compared with other time series tasks, the characteristic of forecasting tasks is that their labels are all based on the time series itself, which can be severely affected by missing values and anomalies [1]. Therefore, the relationship between forecasting tasks and time series imputation tasks is highly close, and this requires attention. In response to such situations, we propose a new perspective of evaluating imputation methods by utilizing the impact of label imputation on downstream forecasting tasks and providing practical tools for application. Due to the importance and widespread application of time series forecasting tasks [2], we believe that we have sufficient motivation to incorporate the effectiveness of time series forecasting into the evaluation of time series imputation. As for other kinds of tasks, we can conduct further research in the future.
>
> >Weakness 2
>
> We fully agree with the value of evaluating the accuracy of imputation as you mentioned. However, we need to emphasize that compared to direct evaluation, evaluating through downstream forecasting tasks has universality and irreplaceability. Universality refers to the fact that unlike time series forecasting tasks, which can obtain true labels through future observations, imputation tasks cannot obtain true labels in reality. Therefore, comparing the accuracy of imputation can only be done on simulated datasets, which limits the objects of evaluation. Irreplaceability refers to the absence of a causal relationship between the accuracy of imputation and the accuracy of downstream forecasting tasks, therefore they cannot replace each other but complement. One of the contributions of our paper is to propose this viewpoint and provide a practical calculation method for evaluation. More innovative time series imputation evaluation methods are also one of the current focuses in the field [3].
>
> In addition, we have two points that need to be clarified, the first is whether our method may harm the accuracy of imputation. Because forecasting models rely on capturing trends, periods, and other characteristics of time series, which in turn has a supervisory effect on time series imputation. Therefore, our method is unlikely to cause significant damage to time series imputation. (shown in the table below).
>
> | MSE(imputation) |    ELE    |
> |:---------------:|:---------:|
> |      SAITS      |   0.398   |
> |      BRITS      |   0.512   |
> |      GPVAE      | 0.493     |
> |      USGAN      | 0.488     |
> |        IT       | 0.358     |
> |       SPIN      | 0.522     |
> |    SAITS+ours   | **0.343** |
> |    BRITS+ours   | 0.368     |
> |    GPVAE+ours   | 0.364     |
> |    USGAN+ours   | 0.368     |
> |     IT+ours     | 0.347     |
> |       SPIN+ours      | 0.352     |
>
> And then the second is that a better imputation accuracy does not always mean a better forecasting performance. We simulate a dataset based on the GEF dataset to illustrate this viewpoint. We conduct experiments on GEF with a predicted length of 24, that is, the label length was 24. We suppose that we only observed the value at the time step nk(k$\geq$0) and nk+1(k$\geq$1). Note that this setting is for the convenience of linear interpolation. In the first case(represented by I), we set n = 4, fill the missing value with linear interpolation, and uniformly add Gaussian noise $\mathcal{N}$(0.05,0.3). In the second case(represented by II), we set n=6 and only do the linear interpolation (visualization can be seen in the pdf uploaded).  We put two data sets into MLP and calculated the forecasting error as shown in the following table.
>
> | MSE | Imputation | Forecasting |
> |:---:|:----------:|:-----------:|
> |  I  | 0.1039 |   **0.1140**  |
> |  II |  **0.0576** |  0.1395 |
>
> This result indicates that there is no causal relationship between imputation accuracy and forecasting accuracy. When we focus on the impact on downstream forecasting tasks, directly focusing on the accuracy of downstream tasks would be a better choice.
>
> >Weakness 3
>
> We highly appreciate the value of the robust forecasting method you mentioned and we have a brief discussion
> about conclusions in robust forecasting methods like [1] in the introduction. Here we want to clarify that our method and robust forecasting methods such as RobustTSF[1] are not in a competitive relationship but in a cooperative relationship, as our method can provide them with data with less noise and higher information data. Specifically, the table below shows the experimental results on the ELE dataset, which indicate that our method can further improve forecasting accuracy based on robust forecasting methods.
>
> |       MSE       |  Mean | SAITS | BRITS | USGAN | GPVAE | ImputerFormer |  SPIN |
> |:---------------:|:-----:|:-----:|:-----:|:-----:|:-----:|:-------------:|:-----:|
> |    +RobustTSF   | 0.056 | 0.050 | 0.092 | 0.084 | 0.099 |     0.053     | 0.076 |
> | +RobustTSF+ours |   -   | **0.046** | 0.050 | 0.052 |**0.046**  |     0.048     | 0.051 |
>
> [1] Cheng H, Wen Q, Liu Y, et al. RobustTSF: Towards Theory and Design of Robust Time Series Forecasting with Anomalies[C]//The Twelfth International Conference on Learning Representations.
>
> [2] Lim B, Zohren S. Time-series forecasting with deep learning: a survey[J]. Philosophical Transactions of the Royal Society A, 2021, 379(2194): 20200209.
>
> [3] Wang J, Du W, Cao W, et al. Deep learning for multivariate time series imputation: A survey[J]. arXiv preprint arXiv:2402.04059, 2024.

---

> > ### Author Response · Authors · 2024-08-11
> > **Kindly Request for Reviewer's Feedback**
> >
> > Dear Reviewer,
> >
> > Since the End of author/reviewer discussions is coming soon, may we know if our response addresses your main concerns? If so, we kindly ask for your reconsideration of the score. Should you have any further advice on the paper and/or our rebuttal, please let us know and we will be more than happy to engage in more discussion and paper improvements.
> >
> > Thank you so much for devoting time to improving our paper!

---

> > > ### Comment · Reviewer_z2pu · 2024-08-12
> > >
> > > Thanks for the responses. Since some concerns remain unsolved, e.g., applications and baselines in experiments, I will keep the score.

---

> ### Author Response · Authors · 2024-08-12
> **Response to reviewer z2pu**
>
> Dear reviewer z2pu
>
> Thank you very much for your feedback. Here we would like to make a simple clarification just in case of any missing.
>
> >Regarding the baseline
>
> We have added the comparison with a robust forecasting baseline called RobustTSF. The results show that the relationship between our approach and them is not competitive, but cooperative. That is, we can help them further obtain better forecasting results.
>
> | MSE | Mean | SAITS | BRITS | USGAN | GPVAE | ImputerFormer | SPIN |
> | :--- | :---: | :---: | :---: | :---: | :---: | :---: | :---: |
> | +RobustTSF | 0.056 | 0.050 | 0.092 | 0.084 | 0.099 | 0.053 | 0.076 |
> | +RobustTSF+ours | - | **0.046** | 0.050 | 0.052 | **0.046** | 0.048 | 0.051 |
>
> >Regarding the application
>
> Our main contribution is to propose utilizing forecasting performance to adjust imputation methods. We believe that time series forecasting is an important task, therefore it is necessary to propose specialized methods for it. As we mentioned in the introduction, it is difficult to find a perfect method that is applicable to all scenarios[1]. Combining different excellent methods for different scenarios is a possible strategy to solve this situation. Therefore, thank you very much for your suggestion. In future work, we will also consider designing corresponding strategies for different types of tasks.
>
> Thank you again for the kind feedback.
>
> [1] Wang J, Du W, Cao W, et al. Deep learning for multivariate time series imputation: A survey[J]. arXiv preprint arXiv:2402.04059, 2024.

---

### Official Review · Reviewer_wGbj · 2024-07-14

**Soundness:** 3
**Presentation:** 2
**Contribution:** 2
**Rating:** 5
**Confidence:** 4

**Summary:**

This paper presents a novel strategy to evaluate time series imputation methods based on their impact on downstream tasks, without requiring multiple model retrainings. The proposed method leverages a similarity calculation to estimate the effect of imputed values efficiently, balancing performance and computational cost. Furthermore, the authors introduce a framework that combines various imputation strategies to enhance downstream task performance, demonstrating significant improvements in forecasting accuracy.

**Strengths:**

1. Unlike traditional methods that focus solely on data restoration, this paper shifts the evaluation criterion to the performance of imputed data in subsequent forecasting tasks. This innovative perspective addresses a significant gap in the field and offers a more practical assessment of imputation quality.

2. The research is supported by thorough theoretical analysis and experiments on multiple benchmark datasets. The authors provide clear justifications for their approach and present detailed comparisons with existing methods.

**Weaknesses:**

1. Although the paper introduces a retrain-free method to estimate the impact of missing labels, it does not thoroughly discuss the computational efficiency of this approach compared to traditional methods, particularly for large-scale datasets.

2. The motivation of this paper is not very clear: why considering the impact of missing values as labels in downstream forecasting tasks is a valuable formulation, and why is this the optimal way to combine existing advanced methods

**Questions:**

1. Is the proposed method a post-processing evaluation or an in-processing method during model training?
2. Is the method only comparing one pair of models at a time?
3. What is the approximation error of the indicator function and how does this impact the imputation performance?

**Limitations:**

Yes

---

> ### Author Rebuttal · Authors · 2024-08-06
>
> We thank the reviewer for the valuable feedback and we have addressed each of the concerns raised by the reviewer as outlined below.
>
> >Weakness 1
>
> To our knowledge, we are the first to examine the impact of missing values as training labels on downstream forecasting tasks, so there is no absolutely fitting traditional method. In section 3.3.1 and Appendix C.4, we modified the Influence function(IF) [1] and made it applicable to our tasks. Theoretically, the complexity of our method is O (pL), where p is the number of network parameters and L is the predicted length. If further optimization is not considered, the complexity of IF will be O ($p ^ 3$+pL). If the L-BFGS algorithm is used to approximate the inverse of the Hessian matrix [1], the complexity of IF will also be O (pL). However, with the accelerated calculation method in Section 2.3 using the properties of time series, our approach can achieve less computation time than IF and outperforms it.
>
> For large-scale data, we applied our method to a 15-minute resolution UCI electricity dataset (with approximately 100000 training points). To make the experiment even more complex, we adjusted our experimental setup to input 96 points and output 96 points, and here is the result.
>
> |   ELE   |  Mean | SAITS | Mean+SAITS+IF | Mean+SAITS+ours | Mean+SAITS+ours_seg16 | Mean+SAITS+ours_seg8 |
> |:-------:|:-----:|:-----:|:-------------:|:---------------:|:---------------------:|:--------------------:|
> |   MSE   | 0.249 | 0.307 |     0.248     |      0.238      |       **0.236**       |         0.237        |
> | Time(s) |   -   |   -   |     126.33    |     1053.28     |         264.27        |      **118.53**      |
>
> It can be seen that the computational complexity is acceptable in practice.
>
> >Weakness 2
>
> We first need to point out that imputation tasks are usually not the goal, but often run as a preprocessing part, and the processed time series needs to be applied to various types of downstream tasks. Evaluating imputation strategies based on the performance of downstream tasks has received increasing attention[1] and the time series forecasting task is an important component of time series-related tasks. Although this may not necessarily be the best solution, establishing an end-to-end method for filling data applied to time series forecasting is valuable. In addition, we want to point out that better imputation accuracy does not always mean better forecasting performance. We simulate a dataset based on the GEF dataset to illustrate this viewpoint and conduct an experiment with a predicted length of 24. We suppose that we only observed the value at the time step nk(k$\geq$0) and nk+1(k$\geq$1). Note that this setting is for the convenience of linear interpolation. In the first case(represented by I), we set n = 4, fill the missing value with linear interpolation, and uniformly add Gaussian noise $\mathcal{N}$(0.05,0.3). In the second case(represented by II), we set n=6 and only do the linear interpolation (visualization can be seen in the pdf uploaded). We put two data sets into MLP and calculated the forecasting error as shown in the following table.
>
> | MSE | Imputation | Forecasting |
> |:---:|:----------:|:-----------:|
> |  I  | 0.1039 |   **0.1140**  |
> |  II |  **0.0576** |  0.1395 |
>
> This result indicates that there is no causal relationship between imputation accuracy and forecasting accuracy. When we focus on the impact on downstream forecasting tasks, directly focusing on the accuracy of downstream tasks would be a better choice.
>
> >Question 1
>
> Our method can calculate both after training and during training (note that it is the model in the downstream forecasting model rather than the imputation model). The difference between them is whether we examine the parameters at every moment during the entire training process or only focus on the parameters after the training is completed. In our experiment, we performed calculations synchronously during the training of downstream forecasting models.
>
> >Question 2
>
> Yes, our method mainly involves pairwise comparisons. In fact, from Tables 2 and 3 in the original paper, it can be seen that a simple pairwise comparison can achieve better results than the original (imputation) model.
>
> >Question 3
>
> We can analyze the approximation error from two parts.
>
> Firstly, in Section 2, we first approximated the effect
> of the label $y_{i,l}$ with a first-order term. This is reasonable for the MSE loss function since the second-order term doesn't affect the model parameters and thus the test set performance($\frac {\partial^2\frac{\partial \mathcal{L}(f_\theta(x),y)}{\partial \theta }}{\partial y_{i,l}^2} = 0$). Furthermore, Lemma1 shows that with a sufficiently small fitting error, the derivatives' fitting error can also be small with any probability $p$. Notably, in practice, we only need to determine if the imputation is beneficial at each step, focusing on positive or negative gain. This implies that our estimation doesn't need high accuracy yet still brings significant gains to downstream tasks (see experiments in section 3.2.2).
>
> Secondly, the above analysis indicates that our fitting error depends on the error of using the NTK kernel function to fit the neural network. It is difficult to directly analyze this error. However, many related works have demonstrated the rationality of doing so through extensive experiments[2,3].
>
>
> [1] Wang J, Du W, Cao W, et al. Deep learning for multivariate time series imputation: A survey[J]. arXiv preprint arXiv:2402.04059, 2024.
>
> [2] Jacot A, Gabriel F, Hongler C. Neural tangent kernel: Convergence and generalization in neural networks[J]. Advances in neural information processing systems, 2018, 31.
>
> [3] Tsai C P, Yeh C K, Ravikumar P. Sample-based explanations via generalized representers[J]. Advances in Neural Information Processing Systems, 2024, 36.

---

> > ### Comment · Reviewer_wGbj · 2024-08-10
> > **Response to author rebuttal**
> >
> > Thanks to the authors for the detailed response! I have gone through the rebuttal and most of my concerns have been addressed. Based on the new results, it is evident that the pairwise model comparison might increase the computational complexity of the overall procedure, given that the authors claim the computational complexity is within an acceptable range. I will keep my rating.

---

> ### Author Response · Authors · 2024-08-12
> **Response to reviewer wGbj**
>
> Dear reviewer wGbj
>
> Thank you very much for your feedback, and we are also very pleased that our response can address most of your concerns. We want to make a small addition regarding the additional time burden just in case of misunderstanding. We list the time for applying our method to obtain the result in Table 2 of the original paper, as shown in the table below.
>
> As the table below shows, our method only requires about 30 seconds of computation time on the ELE dataset. For imputation methods like SAITS (which performs well in our comparison methods), the time required for imputation will exceed 1500 seconds. This means that our additional computational burden will be less than **2%**.
>
> |         |   GEF  |  ETTH1 |  ETTH2 |   ELE  | Traffic | AIR    |
> |:-------:|:------:|:------:|:------:|:------:|:-------:|--------|
> | Time(s) | 19.908 | 20.233 | 20.908 | 30.170 |  29.132 | 14.701 |
>
> Thank you again for your kind feedback.

---

### Official Review · Reviewer_aC85 · 2024-07-15

**Soundness:** 3
**Presentation:** 2
**Contribution:** 3
**Rating:** 5
**Confidence:** 3

**Summary:**

This paper focuses on the imputation of missing values in time series. By noticing that different imputation methods might affect the downstream forecasting tasks, this paper proposes the imputation evaluation approach regarding the downstream tasks' performance.
Then, the authors also developed some methods to improve the efficiency of the evaluation approach and did experiments against some baselines.

**Strengths:**

1. This paper proposed an evaluation approach to time series imputation methods from the perspective of the impact on the downstream forecasting task. This is a seemingly interesting perspective.

2. The proposed method has mathematically formulated the intuition of evaluating imputation methods based on the prediction performance, and the authors developed the approximate method to improve the efficiency.

**Weaknesses:**

1. The evaluation criteria defined in (1) are based on binary comparison and do not look realistic. In real-world applications, the difference in prediction errors between the two methods can be marginal and insignificant, and calculating this binary indicator makes little sense. Meanwhile, in real-world data, the percent of missing values would be very low, and it is doubtful that imputed missing values would lead to significant differences in downstream predictions, especially considering that there are various techniques to regularize models for more robust predictions.

2. Some mathematical presentation is confusing. For instance, how is Eq.(2) derived?  $f(X_k^v, \theta)$ is irrelevant to $y$, and thus why is there the term of the derivative of  $f(X_k^v, \theta)$ w.r.t. $y$ in Eq.(2)

   According to line 148,  Eq.(3) is for approximating $\frac{\partial f(X_k^v, \theta)}{y_{i, l}}$, while on line 156, it presents the substitute of $\frac{\partial f(X_k^v, \theta)}{y_{i, l}}$ again. This is very confusing.

3. The experiments have unjustified and confusing setups and results. For instance, the description between lines 259 and 267 is very unclear, e.g., what is the label in a time series prediction task? why specifically replace 40% of data?
In Fig. 1, what is the percentage w.r.t. on the x-axis?

**Questions:**

See above.

**Limitations:**

No potential negative societal impact is found in this work.

---

> ### Author Rebuttal · Authors · 2024-08-06
>
> We thank the reviewer for the valuable feedback and we will address each of the concerns raised by the reviewer as outlined below.
> >Weakness 1
>
> Firstly, we would like to clarify the importance and necessity of handling missing values even though there is just a small difference. The phenomenon of missing time series is widely present in fields such as power systems[1] and healthcare[2] and the missing rate might not be low in the real world. For example, the missing rate in PhysioNet2012[2] might be as high as 80%. The missing values in the power system can affect downstream tasks such as load forecasting and power dispatch, where a small deviation is enough to cause huge economic losses. In healthcare, such deviations are closely related to the lives of patients. With a responsible attitude towards people's lives and property, we believe that even minor deviations are worth paying attention to, not limited to these areas.
>
> Secondly, we highly appreciate the value of the robust forecasting methods mentioned. However, these methods are not contradictory to our approach but can be combined. We applied our method to the latest related work RobustTSF[3], and the results on the ELE dataset are shown in the table below.
>
> |       MSE       |  Mean | SAITS | BRITS | USGAN | GPVAE | ImputerFormer |  SPIN |
> |:---------------:|:-----:|:-----:|:-----:|:-----:|:-----:|:-------------:|:-----:|
> |    +RobustTSF   | 0.056 | 0.050 | 0.092 | 0.084 | 0.099 |     0.053     | 0.076 |
> | +RobustTSF+ours |   -   | **0.046** | 0.050 | 0.052 |**0.046**  |     0.048     | 0.051 |
>
> From the results, it can be seen that our method can help further improve the accuracy of robust forecasting.
>
> >Weakness 2
>
> We are very sorry for the misunderstanding caused.
>
> 1. The symbolic expression $\frac{\partial f\left(X_k^v, \theta\right)}{\partial y_{i,l}}$ can be conceptually broken down into $\frac{\partial f\left(X_k^v, \theta\right)}{\partial \theta} \cdot \frac{\partial \theta}{\partial y_{i,l}}$, elucidating the role of the label $y_{i,l}$ in shaping the model parameters $\theta$ throughout the training process. This, in turn, has repercussions on the model's prediction when evaluated on unseen data from the test set, i.e. $f\left(X_k^v, \theta\right)$. By focusing on the derivative $\frac{\partial f\left(X_k^v, \theta\right)}{\partial y_{i,l}}$, our goal is to assess the extent to which changes in label values $y_{i,l}$ influence the model's predictions on the test set, thereby affecting overall model efficacy.
>
> 2. Your understanding of Eq.(3) as an approximation for the partial derivative $\frac{\partial f\left(X_k^v, \theta\right)}{\partial y_{i, l}}$, where $X_k^v$ represents samples from the test set, is correct. When we aim to approximate the derivatives for the test set using Eq.(3), we essentially leverage information obtained from the training set. This involves utilizing the derivatives $\frac{\partial f\left(X_j, \boldsymbol{\theta}\right)}{\partial y_{i, l}}$ calculated at various points $X_j$ within the training set to construct or "fit" a function that can then estimate these derivatives for any given point in the test set.
>
> As for other content that may cause misunderstandings, we will carefully review and revise it in the revised version of the paper.
>
> >Weakness 3
>
> We are very sorry for the misunderstanding.
>
> 1. We detailed the experimental setup in section 3.1 of our paper. The label represents the values of the next 24 data points following the current time point. Given that our data is recorded at an hourly resolution, this allows us to predict the time series for the full 24 hours ahead.
>
> 2. The missing rate of 40% is a widely used experimental setting[4]. Due to time and computational resource limitations (mainly from the imputation method itself rather than our method), in the following, we only provide the experimental results with a missing rate of 50%. The results again show the superiority of our method. We will include more results with various missing rates in our revised paper.
>
> |       MSE in ELE(50% missing)       |  Mean | SAITS | BRITS | USGAN | GPVAE | ImputerFormer |
> |:---------------:|:-----:|:-----:|:-----:|:-----:|:-----:|:-------------:|
> |      -    | 0.141 | 0.168 | 0.208 | 0.207| 0.204 |     0.134     |
> | +ours |   -   | **0.112** | 0.131 | 0.131 |0.128  |     0.115     |
>
> 3. For the x-axis in Fig. 1, we described it in the caption of Fig. 1 in the original paper, the meaning here is that we selected the time step with the highest absolute value in gain estimation, which is the top x% time step that impacts most after the imputation, and compared the accuracy and correlation between our method's estimation and the actual gain. This kind of evaluation is widely used in related work [5].
>
> [1] Park J Y, Nagy Z, Raftery P, et al. The Building Data Genome Project 2, energy meter data from the ASHRAE Great Energy Predictor III competition[J].
>
> [2] Silva I, Moody G, Scott D J, et al. Predicting in-hospital mortality of ICU patients: The physionet/computing in cardiology challenge 2012[C]//2012 computing in cardiology. IEEE, 2012: 245-248.
>
> [3] Cheng H, Wen Q, Liu Y, et al. RobustTSF: Towards Theory and Design of Robust Time Series Forecasting with Anomalies[C]//The Twelfth International Conference on Learning Representations.
>
> [4] Du W, Côté D, Liu Y. Saits: Self-attention-based imputation for time series[J]. Expert Systems with Applications, 2023, 219: 119619.
>
> [5] Koh P W, Liang P. Understanding black-box predictions via influence functions[C]//International conference on machine learning. PMLR, 2017: 1885-1894.

---

> > ### Author Response · Authors · 2024-08-12
> > **Kindly Request for Reviewer's Feedback**
> >
> > Dear Reviewer aC85,
> >
> > Since the End of author/reviewer discussions is coming in ONE day, may we know if our response addresses your main concerns? If so, we kindly ask for your reconsideration of the score. Should you have any further advice on the paper and/or our rebuttal, please let us know and we will be more than happy to engage in more discussion and paper improvements.
> >
> > Thank you so much for devoting time to improving our paper!

---

> > > ### Comment · Reviewer_aC85 · 2024-08-12
> > >
> > > Sorry for the late response. Thanks for the authors' answers.
> > >
> > > Most of my questions are addressed. I will update the rating accordingly.

---

> ### Author Response · Authors · 2024-08-14
> **Kindly Remind for the Coming Ending of Discussion Period**
>
> Dear Reviewer aC85,
>
> Thank you very much for your recognition of our response and encouragement of our work. We are very pleased that our response can answer most of your questions. We noticed that you mentioned updating the rating for our work. As the discussion period is coming to an end, could you please take the time to update the rating?
>
> If you have any further questions about our work, we are delighted to continue discussing with you to alleviate your concerns, which will greatly help improve the quality of our work.

---

### Author Rebuttal · Authors · 2024-08-06

We sincerely thank all the reviewers for their valuable time and detailed comments, and we appreciate that the reviewer recognized the strength of our paper like the **theoretical analysis**(Reviewer **aC85**, **wGbj**), **innovative perspective**(Reviewer **wGbj**, **56k9**, **xbnY**), and **practical value**(Reviewer **z2pu**). We have carefully read the comments and provide a detailed response to each comment below. Here we highlight our major responses to each reviewer below. We hope our responses can properly address your concerns.

* In response to the comments of Reviewer **xbnY**'s feedback that we should add more up-to-date imputation methods for comparison, we add two of the SOTA methods called Imputeformer and SPIN and use them to strengthen Table 2 in the original paper(reviewers can find it in the attached pdf). The experiment shows that our methods can help improve even the most up-to-date methods.

* Regarding the concerns of the reviewers **aC85** and **z2pu** about whether there is an overlap between our method and the robust forecasting method. We clarified that our relationship with the robust forecasting method is not a competition but a win-win cooperation, and added experiments showing that our method can further improve the accuracy of robust forecasting.

* We demonstrated to reviewers **56K9** and **xbnY** the scalability of our method, which can be extended to multivariate time series forecasting and also to examine the impact of input data $X$.

* To address the concerns of reviewers **wGbj** and **56K9** about the computational efficiency of our method, we provided the computation time of our method and validated it on a larger scale of data, demonstrating the computational efficiency of our method and the effectiveness of the acceleration method mentioned in section 2.3 of the original paper.

* To better explain to reviewer **z2pu** and **wGbj** the necessity of incorporating the performance of downstream forecasting tasks into the evaluation of time series imputation, we constructed a simulated dataset (relevant visualizations can be found in the PDF file we uploaded) to demonstrate that there is no causal relationship between higher time imputation accuracy and higher forecasting accuracy. When we focus on the impact on downstream forecasting tasks, directly focusing on the accuracy of downstream tasks would be a better choice.

---

### Decision · Program_Chairs · 2024-09-25

**Decision:**

Accept (poster)

**Comment:**

# Summary
This paper proposes a novel approach to evaluating time series imputation methods, focusing on their impact on downstream forecasting tasks rather than just the accuracy of the imputed data. The proposed approach efficiently estimates the effect of different imputation strategies using a similarity-based method, avoiding the need for multiple model retrainings.   Additionally, the authors present a framework that combines the strengths of various imputation techniques to enhance overall task performance, demonstrating significant improvements in downstream forecasting tasks.

The strengths of the paper include its innovative perspective, strong theoretical foundation, and practical relevance. It addresses a gap in the field by providing a more meaningful assessment of imputation quality in real-world applications.

# Concerns and Author Feedback
However, there are concerns about unclear mathematical explanations, limited scope (focusing only on forecasting), and the computational efficiency of the proposed method. Some reviewers also noted that the paper lacks comparisons with the latest imputation techniques and questioned its generalizability to other tasks and multivariate data.

In their feedback, the authors clarified several points, added comparisons with state-of-the-art methods, took steps toward demonstrating the scalability and efficiency of their approach, and provided limited results for the multivariate setting showing consistent improvements as with univariate. They also provided additional experiments to address concerns about the necessity of focusing on downstream tasks.

# Pros and Cons
- Pros:
  - Innovative approach that shifts the evaluation of imputation methods to their impact on downstream tasks.
  - Strong theoretical foundation and practical relevance.
  - Demonstrated forecast performance improvements and scalability.
- Cons:
  - Mathematical explanations and some experimental setups are unclear.
  - Limited scope, focusing only on forecasting tasks.
  - Concerns about computational efficiency and lack of comparisons with state-of-the-art imputation methods.
  - Questions about the generalizability to other time series tasks and multivariate data.

# Final ratings
After rebuttal and discussions 4/5 reviewers were on the accept side, with one full accept, all others including reject on the borderline, with various different shortcomings listed above being the reason for maintaining borderline ratings. This is a majority on the accept side with at least one feeling more strongly than borderline for acceptance.

# Final Decision Justification:
The decision to accept the paper is based on its innovative and practically valuable approach, which outweighs the concerns raised. The authors' responses and additional experiments further strengthen the paper's contributions, making it a worthwhile addition to the field.